# mRNA vaccine-induced T cells respond identically to SARS-CoV-2 variants of concern but differ in longevity and homing properties depending on prior infection status

Jason Neidleman[1,2†], Xiaoyu Luo[2†], Matthew McGregor[1,2], Guorui Xie[1,2], Victoria Murray[3], Warner C Greene[2], Sulggi A Lee[4*], Nadia R Roan[1,2*]

[1]Department or Urology, University of California, San Francisco, San Francisco, United States; [2]Gladstone Institute of Virology, San Francisco, United States; [3]University of California, San Francisco, San Francisco, United States; [4]Medicine, University of California, San Francisco, San Francisco, United States

*For correspondence:
Sulggi.lee@ucsf.edu (SAL);
nadia.roan@gladstone.ucsf.edu (NRR)

†These authors contributed equally to this work

Competing interest: The authors declare that no competing interests exist.

**ABSTRACT** While mRNA vaccines are proving highly efficacious against SARS-CoV-2, it is important to determine how booster doses and prior infection influence the immune defense they elicit, and whether they protect against variants. Focusing on the T cell response, we conducted a longitudinal study of infection-naïve and COVID-19 convalescent donors before vaccination and after their first and second vaccine doses, using a high-parameter CyTOF analysis to phenotype their SARS-CoV-2-specific T cells. Vaccine-elicited spike-specific T cells responded similarly to stimulation by spike epitopes from the ancestral, B.1.1.7 and B.1.351 variant strains, both in terms of cell numbers and phenotypes. In infection-naïve individuals, the second dose boosted the quantity and altered the phenotypic properties of SARS-CoV-2-specific T cells, while in convalescents the second dose changed neither. Spike-specific T cells from convalescent vaccinees differed strikingly from those of infection-naïve vaccinees, with phenotypic features suggesting superior long-term persistence and ability to home to the respiratory tract including the nasopharynx. These results provide reassurance that vaccine-elicited T cells respond robustly to emerging viral variants, confirm that convalescents may not need a second vaccine dose, and suggest that vaccinated convalescents may have more persistent nasopharynx-homing SARS-CoV-2-specific T cells compared to their infection-naïve counterparts.

## Introduction

A year and a half since the December 2019 emergence of SARS-CoV-2, the novel betacoronavirus had already infected almost 200 million people and taken the lives of over 4 million, nearly collapsed worldwide health systems, disrupted the global economy, and perturbed society and public health on a scale not experienced within the past 100 years. Fortunately, multiple highly efficacious vaccines, including the two-dose mRNA-based ones developed by Pfizer/BioNTech and Moderna, which confer ~90 % protection against disease, were approved for emergency use before the end of 2020. Although the vaccines provide the most promising route for a rapid exit from the COVID-19 pandemic, concerns remain regarding the durability of the immunity elicited by these vaccines and the extent to which they will protect against the variants of SARS-CoV-2 now spreading rapidly around the world.

**eLife digest** Vaccination is one of the best ways to prevent severe COVID-19. Two doses of mRNA vaccine protect against serious illness caused by the coronavirus SARS-CoV-2. They do this, in part, by encouraging the immune system to make specialised proteins known as antibodies that recognise the virus. Most of the vaccine research so far has focussed on these antibodies, but they are only one part of the immune response. Vaccines also activate immune cells called T cells. These cells have two main roles, coordinating the immune response and killing cells infected with viruses. It is likely that they play a key role in preventing severe COVID-19.

There are many kinds of T cells, each with a different role. Currently, the identity and characteristics of the T cells that protect against COVID-19 is unclear. Different types of T cells have unique proteins on their surface. Examining these proteins can reveal details about how the T cells work, which part of the virus they recognise, and which part of the body they protect. A tool called cytometry by time of flight allows researchers to measure these proteins, one cell at a time.

Using this technique, Neidleman, Luo et al. investigated T cells from 11 people before vaccination and after their first and second doses. Five people had never had COVID-19 before, and six had already recovered from COVID-19. Neidleman, Luo et al. found that the T cells recognizing SARS-CoV-2 in the two groups differed. In people who had never had COVID-19 before, the second dose of vaccine improved the quality and quantity of the T cells. The same was not true for people who had already recovered from COVID-19. However, although their T cells did not improve further after a second vaccine dose, they did show signs that they might offer more protection overall. The proteins on the cells suggest that they might last longer, and that they might specifically protect the nose, throat and lungs. Neidleman, Luo et al. also found that, for both groups, T cells activated by vaccination responded in the same way to different variants of the virus.

This work highlights the importance of getting both vaccine doses for people who have never had COVID-19. It also suggests that vaccination in people who have had COVID-19 may generate better T cells. Larger studies could show whether these patterns remain true across the wider population. If so, it is possible that delivering vaccines to the nose or throat could boost immunity by mimicking natural infection. This might encourage T cells to make the surface proteins that allow them to home to these areas.

The first variant observed to display a survival advantage was the D614G, which was more transmissible than the original strain and quickly became the dominant variant throughout the world *Korber et al., 2020*. This variant, fortunately, did not evade immunity and in fact appeared to be more sensitive than the original strain to antibody neutralization by convalescent sera *Weissman et al., 2021*. More worrisome, however, was the emergence at the end of 2020 of rapidly spreading variants in multiple parts of the world, including B.1.1.7, B.1.351, P.1, and B.1.427/B.1.429 (originally identified in United Kingdom, South Africa, Brazil, and California, respectively) *Plante et al., 2021*, followed by additional highly transmissible variants in 2021 including the B.1.617.2 which was first detected in India *Callaway, 2021*. Some variants, including B.1.1.7, may be more virulent *Davies et al., 2021*. While antibodies against the original strain elicited by either vaccination or infection generally remain potent against B.1.1.7, their activity against B.1.351 and P.1 is compromised *Wang et al., 2021*; *The CITIID-NIHR BioResource COVID-19 Collaboration et al., 2021*; *Muik et al., 2021*; *Garcia-Beltran et al., 2021*; *Stamatatos et al., 2021*; *Cele et al., 2021*; *Hoffmann et al., 2021*; *Planas et al., 2021*; *Edara et al., 2021*; *Kuzmina et al., 2021*. Antibodies from vaccinees were 14-fold less effective against B.1.351 than against the ancestral strain, and a subset of individuals completely lacked neutralizing antibody activity against B.1.351 9 months or more after convalescence *Planas et al., 2021*.

Reassuringly, early data suggest that relative to antibody responses, T-cell-mediated immunity appears to be less prone to evasion by the variants *Skelly et al., 2021*; *Tarke et al., 2021*; *Redd et al., 2021*; *Geers et al., 2021*; *Woldemeskel et al., 2021*; *Stankov et al., 2021*; *Tauzin et al., 2021*. Among 280 CD4+ and 523 CD8+ T cell epitopes from the original SARS-CoV-2, an average of 91.5 % (for CD4) and 98.1 % (for CD8) mapped to regions not mutated in the B.1.1.7, B1.351, P.1, and B.1.427/B.1.429 variants. Focusing on just the spike response, the sole SARS-CoV-2 antigen in the mRNA-based vaccines, then 89.7 % of the CD4+ epitopes and 96.4 % of the CD8+ epitopes

are conserved *Tarke et al., 2021*. In line with this, the magnitude of the response of T cells from convalescent or vaccinated individuals was not markedly reduced when assessed against any of the variants *Tarke et al., 2021*. The relative resistance of T cells against SARS-CoV-2 immune evasion is important in light of the critical role these immune effectors play during COVID-19. T cell numbers display a strong, inverse association with disease severity *Chen et al., 2020*; *Woodruff et al., 2020*, and the frequency of SARS-CoV-2-specific T cells predicts recovery from severe disease *Rydyznski Moderbacher et al., 2020*; *Neidleman et al., 2021*. SARS-CoV-2-specific T cells can also provide long-term, self-renewing immunological memory: these cells are detected more than half a year into convalescence and can proliferate in response to homeostatic signals *Dan et al., 2021*; *Neidleman et al., 2020b*. Furthermore, the ability of individuals with inborn deficiencies in B cell responses to recover from COVID-19 without intensive care suggests that the combination of T cells and innate immune mechanisms is sufficient for recovery when antibodies are lacking *Soresina et al., 2020*.

Although T cells against the ancestral strain display a response of similar magnitude and breadth to the variants *Tarke et al., 2021*, to what extent these T cells' phenotypes and effector functions differ during their response to variant detection is a different question. Small changes in the sequences of T cell epitopes, in the form of altered peptide ligands (APLs), can theoretically alter how the T cells respond to stimulation. Indeed, change of a single residue can convert a proliferative, IL4-secreting effector response into one that continues to produce IL4 in the absence of proliferation *Evavold and Allen, 1991*. Furthermore, APLs can activate Th1 cells without inducing either proliferation or cytokine production, shift Th1 responses into Th2-focused ones, and in some instances even render T cells anergic or immunoregulatory by eliciting TGFβ production *Sloan-Lancaster and Allen, 1996*.

Another important aspect that has not been explored is to what extent vaccine- vs. infection-induced T cell responses differ phenotypically and functionally, and to what extent convalescent individuals benefit from vaccination as they already harbor some form of immunity against the virus. Studies based on the antibody and B cell response suggest that for COVID-19 convalescents, a single dose of the mRNA vaccines is helpful while the additional booster is not necessary *Stamatatos et al., 2021*; *Goel et al., 2021*; *Ebinger et al., 2021*; how this translates in the context of vaccine-elicited T cell immunity is not clear.

To address these knowledge gaps, we conducted 39-parameter phenotyping by CyTOF on 33 longitudinal specimens from 11 mRNA-vaccinated individuals, six of whom had previously contracted and recovered from COVID-19. For each participant, blood specimens were obtained prior to vaccination, two weeks following the first dose, and two weeks following the second. For every specimen, we assessed in depth the phenotypes and effector functions of total CD4+ and CD8+ T cells, and of CD4+ and CD8+ T cells responding to the original SARS-CoV-2 spike, to spike from variants B.1.1.7 and B.1.351, and to nucleocapsid. By conducting analyses on the resulting 165 high-dimensional

**Table 1.** Participant characteristics.

| Patient ID | Gender | Age | Prior infection status | Vaccine | Days post PCR+ test at pre-vaccination timepoint | Days post vaccine dose #1 | Days post vaccine dose #2 |
|---|---|---|---|---|---|---|---|
| PID4101 | Female | 45 | Uninfected | Pfizer/BioNT | NA | 13 | 12 |
| PID4109 | Male | 33 | Uninfected | Pfizer/BioNT | NA | 12 | 33 |
| PID4197 | Female | 76 | Uninfected | Pfizer/BioNT | NA | 14 | 13 |
| PID4198 | Male | 79 | Uninfected | Moderna | NA | 18 | 10 |
| PID4199 | Female | 32 | Uninfected | Pfizer/BioNT | NA | 14 | 10 |
| PID4104 | Female | 33 | Convalescent | Moderna | 212 | 14 | 14 |
| PID4108 | Female | 20 | Convalescent | Pfizer/BioNT | 226 | 13 | 38 |
| PID4112 | Female | 59 | Convalescent | Moderna | 254 | 16 | 13 |
| PID4114 | Female | 46 | Convalescent | Moderna | 216 | 16 | 50 |
| PID4117 | Female | 51 | Convalescent | Pfizer/BioNT | 82 | 16 | 6 |
| PID4118 | Female | 39 | Convalescent | Pfizer/BioNT | 173 | 18 | 28 |

datasets generated, we find a reassuringly unaltered T cell response against the variants, an ability of the booster dose to alter the phenotypes of vaccine-elicited T cells, and a striking impact of prior infection on qualitative features of T cells elicited by vaccination.

## Results
### Study design

To characterize the phenotypic features of mRNA vaccination-elicited SARS-CoV-2-specific T cells, we procured 33 longitudinal blood samples from the COVID-19 Host Immune Response and Pathogenesis (CHIRP) cohort. Four of the participants had received the Moderna (mRNA-1273) vaccine, while the remaining seven had received the Pfizer/BioNTech (BNT162b2) one. For all participants, longitudinal specimens were obtained at three timepoints: prior to vaccination, ~2 weeks (range 13–18 days) after the first vaccine dose, and ~2 weeks (range 6–38 days) after the second dose. Five of the participants were never infected with SARS-CoV-2, while the remaining six had completely recovered from mild (non-hospitalized) COVID-19 disease (*Table 1*). These prior infections had all occurred in the San Francisco Bay Area between March and July of 2020, when the dominant local strain was the original ancestral strain. Each specimen was phenotyped using a 39-parameter T cell-centric CyTOF panel (see Materials and methods and *Table 2*) at baseline (to establish the overall phenotypes of total CD4+ and CD8+ T cells), and following 6 hr of stimulation with overlapping 15-mer peptides spanning the entire original (ancestral) SARS-CoV-2-spike, B.1.1.7 spike, B.1.351 spike, or the original SARS-CoV-2 nucleocapsid (the latter as a control for a SARS-CoV-2-specific response not boosted by vaccination). Including all the baseline and stimulation conditions, a total of 165 specimens from the 11 participants were analyzed by CyTOF.

### SARS-CoV-2-specific T cells elicited by vaccination recognize B.1.1.7 and B.1.351 variants

We first confirmed our ability to identify SARS-CoV-2-specific T cells by stimulating PBMCs from vaccinated individuals with spike peptides. In line with our prior studies implementing a 6 hr peptide stimulation *Neidleman et al., 2021*; *Neidleman et al., 2020b*, spike-specific CD4+ T cells could be specifically identified through intracellular cytokine staining for IFNγ, and a more robust

**Table 2.** List of CyTOF antibodies used in study. Antibodies were either purchased from the indicated vendor or prepared in-house using commercially available MaxPAR conjugation kits per manufacturer's instructions (Fluidigm).

| Antigen target | Clone | Elemental isotope | Vendor |
|---|---|---|---|
| HLADR | TÜ36 | Qdot (112 Cd) | Thermofisher |
| RORγt* | AFKJS-9 | 115 In | In-house |
| CD49d (α4) | 9F10 | 141Pr | Fluidigm |
| CTLA4* | 14D3 | 142Nd | In-house |
| NFAT* | D43B1 | 143Nd | Fluidigm |
| CCR5 | NP6G4 | 144Nd | Fluidigm |
| CD137 | 4B4-1 | 145Nd | In-house |
| CD95 | BX2 | 146Nd | In-house |
| CD7 | CD76B7 | 147Sm | Fluidigm |
| ICOS | C398.4A | 148Nd | Fluidigm |
| Tbet* | 4B10 | 149Sm | In-house |
| IL4* | MP4-25D2 | 150Nd | In-house |
| CD2 | TS1/8 | 151Eu | Fluidigm |
| IL17* | BL168 | 152Sm | In-house |
| CD62L | DREG56 | 153Eu | Fluidigm |
| TIGIT | MBSA43 | 154Sm | Fluidigm |
| CCR6 | 11A9 | 155Gd | In-house |
| IL6* | MQ2-13A5 | 156 Gd | In-house |
| CD8 | RPA-T8 | 157Gd | In-house |
| CD19 | HIB19 | 157Gd | In-house |
| CD14 | M5E2 | 157Gd | In-house |
| OX40 | ACT35 | 158Gd | Fluidigm |
| CCR7 | G043H7 | 159Tb | Fluidigm |
| CD28 | CD28.2 | 160Gd | Fluidigm |
| CD45RO | UCHL1 | 161Dy | In-house |
| CD69 | FN50 | 162Dy | Fluidigm |
| CRTH2 | BM16 | 163Dy | Fluidigm |
| PD-1 | EH12.1 | 164Dy | In-house |
| CD127 | A019D5 | 165Ho | Fluidigm |
| CXCR5 | RF8B2 | 166Er | In-house |
| CD27 | L128 | 167Er | Fluidigm |
| IFNγ* | B27 | 168Er | Fluidigm |
| CD45RA | HI100 | 169Tm | Fluidigm |
| CD3 | UCHT1 | 170Er | Fluidigm |
| CD57 | HNK-1 | 171Yb | In-house |

*Table 2 continued on next page*

*Table 2 continued*

| Antigen target | Clone | Elemental isotope | Vendor |
|---|---|---|---|
| CD38 | HIT2 | 172Yb | Fluidigm |
| α4β7 | Act1 | 173Yb | In-house |
| CD4 | SK3 | 174Yb | Fluidigm |
| CXCR4 | 12G5 | 175Lu | Fluidigm |
| CD25 | M-A251 | 176Yb | In-house |
| CD161 | NKR-P1A | 209 Bi | In-house |

*Intracellular antibodies.

response was observed among CD4+ than CD8+ T cells (*Figure 1* ). No specific induction of IL4 or IL17 by CD4+ T cells were observed in response to peptide stimulation (*Figure 1—figure supplement 1*). In addition, activation-induced markers (AIM) such as Ox40, 4-1BB, and CD69 could also be identified in T cells after spike peptide stimulation, but with a higher background in the baseline (no peptide stimulation) specimens relative to the intracellular cytokine staining approach (*Figure 1—figure supplement 1*). For these reasons, in this study we exclusively used IFNγ positivity in the peptide-stimulated samples as a marker of antigen-specific T cells.

In the infection-naïve participants, the first vaccination dose primed a spike-specific CD4+ T cell response, which was further boosted with the second dose (*Figure 1B*, *top left*). For each participant and time point, similar numbers of cells were stimulated by exposure to the ancestral or variant spikes. This finding suggests that vaccine-elicited spike-specific CD4+ T cells recognize ancestral and variant spike equally well, and is consistent with their recently reported ability to recognize variant strains *Tarke et al., 2021*. The response of vaccine-elicited CD8+ T cells to spike peptides was weaker, and mostly apparent only after the second dose (*Figure 1B*, *top right*). As expected, vaccination did not elicit T cells able to respond to nucleocapsid peptides (*Figure 1C*, *top panels*).

In contrast to the infection-naïve individuals where spike-specific CD4+ T cells were clearly elicited and then boosted upon the second dose, spike-specific CD4+ T cell responses in convalescent individuals did not show a consistent upward trend. Convalescent donor PID4112 had a large frequency of pre-vaccination SARS-CoV-2-specific CD4+ T cells that increased to >1% of the total CD4+ T cell frequency after the first dose and then dampened after dose 2 (*Figure 1B* , *bottom left*). PID4112 also exhibited an elevated nucleocapsid-specific CD4+ T cell response after the first vaccination dose (*Figure 1C*, *bottom left*), which may have been due to bystander effects resulting from the concomitant large spike-specific response. In comparison, PID4112's spike-specific CD8+ T cell response was low after dose 1, and boosted after dose 2 (*Figure 1B*, *bottom right*). In contrast to PID4112, the remaining five convalescent donors exhibited an overall weak spike-specific T cell response. In fact, when comparing these five donors to the five infection-naïve donors, there was a significant decrease in the magnitude of the spike-specific CD4+ T cell response, while the spike-specific CD8+ T cell response was equivalent between the two groups (*Figure 1D*). These results were unexpected and suggest that, when excluding outlier PID4112, the magnitude of the vaccine-elicited spike-specific CD4+ T cell response (after full vaccination) was lower in convalescent individuals than in infection-naïve individuals.

These assessments of the magnitude of the spike-specific T cell response together suggest that (1) in infection-naïve individuals the CD4+ T cell response is boosted by the second vaccination dose, (2) convalescent individuals exhibit a more disparate response, with most donors mounting a weaker response than infection-naïve individuals, and (3) the response is more robust among CD4+ than CD8+ T cells. As a higher number of SARS-CoV-2-specific CD4+ T cells were available for analysis, we focused on this subset for our subsequent analyses.

## Vaccine-elicited spike-specific CD4+ T cells responding to B.1.1.7 and B.1.351 spike are indistinguishable from those responding to ancestral spike

Leveraging our ability to not only assess the magnitude but also the detailed (39-parameter) phenotypic features of SARS-CoV-2-specific CD4+ T cells, we first determined whether the ancestral and variant spike epitopes stimulated different subsets of vaccine-elicited spike-specific CD4+ T cells. Such differences could theoretically result from the fact that ~5–10% of the spike epitopes differ between variants and ancestral strains *Tarke et al., 2021*, and may therefore act as APLs steering responding

## Figure 1

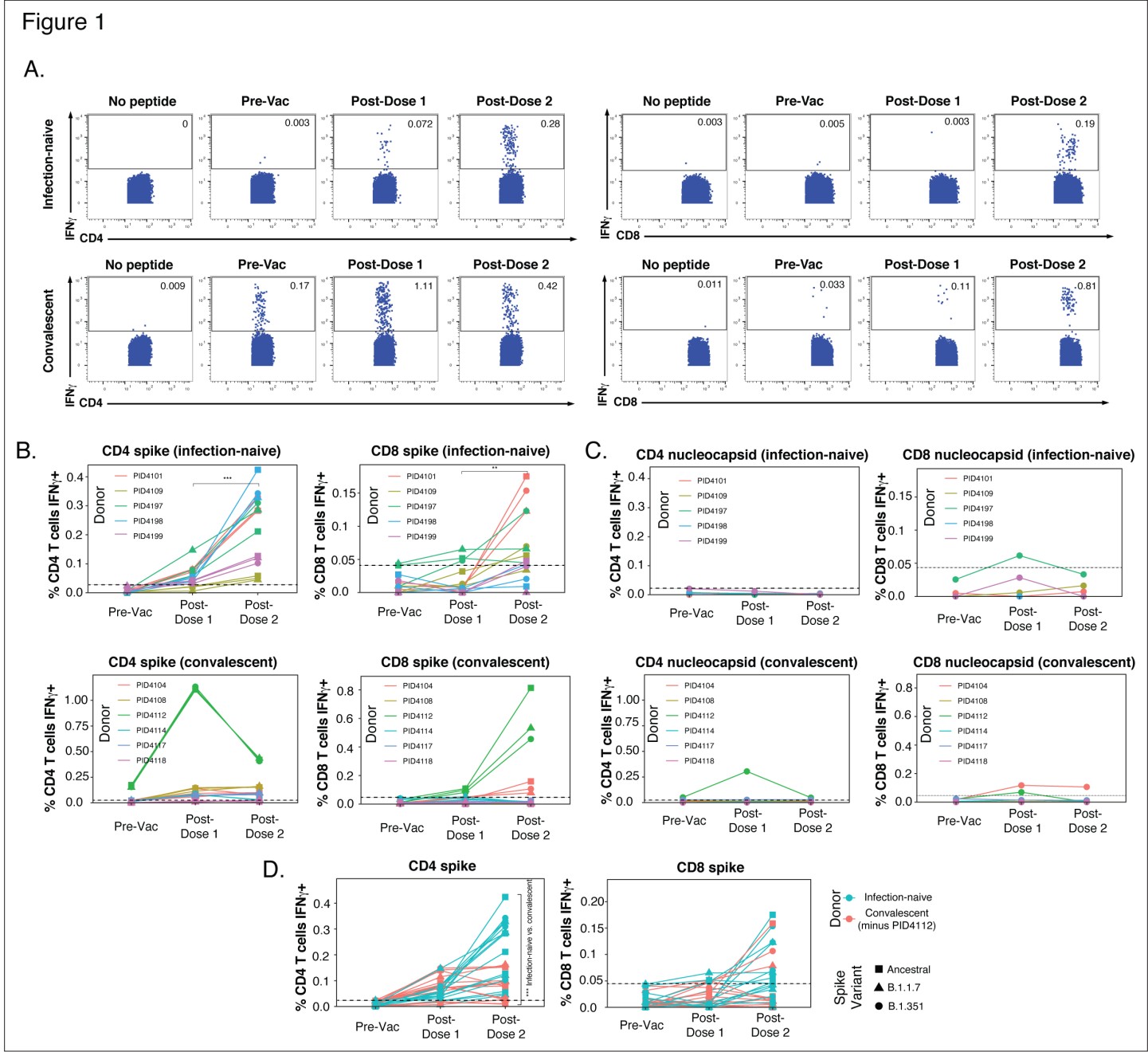

**Figure 1.** SARS-CoV-2-specific T cells elicited by vaccination recognize variants, and in a manner that differs among individuals with prior COVID-19. (**A**) Identification of vaccine-elicited spike-specific T cells. PBMCs before vaccination (Pre-Vac) or 2 weeks after each dose of vaccination were stimulated with spike peptides and assessed by CyTOF 6 hr later for the presence of spike-specific (IFNγ-producing) CD4+ (*left*) or CD8+ (*right*) T cells. The 'no peptide' conditions served as negative controls. Shown are longitudinal data from an infection-naïve (PID4101, *top*) and convalescent (PID4112, *bottom*) individual. (**B**) Quantification of the spike-specific CD4+ (*left*) and CD8+ (*right*) T cells recognizing the ancestral (squares), B.1.1.7 (triangles), and B.1.351 (circles) spike peptides in infection-naïve (*top*) and a convalescent (*bottom*) individuals before and after vaccination. Note the similar frequencies of T cells responding to all three spike proteins in each donor, the clear boosting of spike-specific CD4+ T cell frequencies in infection-naïve but not convalescent individuals, and the overall higher proportion of responding CD4+ than CD8+ T cells. The dotted line corresponds to the magnitude of the maximal pre-vaccination response in infection-naïve individuals and is considered as background. The y-axes are fitted based upon the maximal post-vaccination response values for each patient group and T cell subset. The p-values shown (**p < 0.01, ***p < 0.001) were calculated by student's t-test. (**C**) As expected, nucleocapsid-specific T cell responses are generally low over the course of vaccination, with the exception of convalescent donor PID4112. Shown are the frequencies of nucleocapsid-specific CD4+ (*left*) and CD8+ (*right*) T cells, as measured by IFNγ production upon stimulation with ancestral nucleocapsid peptides, in infection-naïve (*top*) and convalescent (*bottom*) individuals. The dotted line corresponds to the magnitude of the maximal pre-vaccination response in infection-naïve individuals and is considered as the background signal. Y-axes are labeled to match the

*Figure 1 continued on next page*

*Figure 1 continued*

corresponding y-axes for spike-specific T cell responses in *panel B*. (**D**) The CD4+ T cell response is boosted by the second vaccine dose to a greater extent in infection-naïve than convalescents individuals. Shown are the frequencies of spike-specific CD4+ (*left*) and CD8+ (*right*) T cells stimulated by the three spike proteins (squares: ancestral; triangles: B.1.1.7; circles: B.1.351) among the infection-naïve (aqua) and convalescent (coral) donors, after removal of outlier PID4112. ***p < 0.001 comparing the infection-naïve vs. convalescent post-dose two specimens, were calculated using student's t-test.

The online version of this article includes the following figure supplement(s) for figure 1:

**Figure supplement 1.** Six-hr stimulation with spike peptides does not induce significant expression of IL4, IL17, or activation markers in SARS-CoV-2-specific T cells.

cells towards different fates. We isolated the datasets corresponding to both post-vaccination time-points for all eleven donors, and then exported the data corresponding to spike-specific CD4+ T cells (as defined by IFNγ production, *Figure 1* ). After reducing the multidimensional single-cell data for each individual specimen to a two-dimensional datapoint through multidimensional scaling (MDS) *Ritchie et al., 2015*, we observed the ancestral spike-stimulated samples to be interspersed among the B.1.1.7- and B.1.351-responding ones (*Figure 2A*). We then visualized the spike-specific CD4+ T cells at the single-cell level. When visualized alongside total (baseline) CD4+ T cells, spike-specific cells occupied a distinct 'island' defined by high expression of IFNγ (*Figure 2B*), suggesting unique phenotypic features of these cells. To better analyze these spike-responding CD4+ T cells, we visualized them in isolation within a new tSNE which clearly demonstrated complete mixing of the cells stimulated by the ancestral, B.1.1.7, and B.1.351 spike proteins (*Figure 2C*). Almost all the responding cells expressed high levels of CD45RO and low levels of CD45RA (*Figure 2D*), suggesting them to be mostly memory cells. These memory CD4+ T cells included central memory T cells (Tcm), T follicular helper cells (Tfh), and those expressing multiple activation markers (CD38, HLADR, CD69, CD25) and receptors known to direct cells to tissues including the respiratory tract (CXCR4, CCR5, CCR6, CD49d) (*Figure 2E*). The expression levels of these and all other antigens quantitated by CyTOF were not statistically different between CD4+ T cells responding to the three spike proteins (*Figure 2—figure supplement 1*). To confirm the identical phenotypes of the three groups of spike-responding cells, we implemented unbiased clustering by FlowSOM. Spike-stimulated cells were clustered into eight subsets, and no subset was preferentially enriched in any one of the three groups (*Figure 2F*). Together, these data suggest that vaccine-elicited spike-specific CD4+ T cells respond in the same manner to spike epitopes from the ancestral or variant strains, and would probably mount similar responses in vivo to infection by all three virus types.

## Phenotypic alterations of spike-specific CD4+ T cells in infection-naïve recipients after the second vaccine dose

We next took advantage of our longitudinal study design to assess for any changes in the differentiation of spike-specific T cell responses over the course of the 2-dose vaccination. As the data presented above suggested no phenotypic differences between CD4+ T cells responding to the ancestral, B.1.1.7, and B.1.351 spike proteins, our subsequent analyses combined these datasets. We first assessed whether, among infection-naïve individuals, the phenotypes of spike-specific CD4+ T cells were different after the first and second doses. While MDS and tSNE visualizations of the data revealed that the cells from the two timepoints were somewhat interspersed (*Figure 3A-B*), FlowSOM clustering suggested some differences in cluster distribution (*Figure 3C-D*). Direct comparison of the cluster frequencies revealed a cluster (B8) significantly enriched after the first dose, and a different cluster (B5) significantly enriched after the second dose (*Figure 3E*). As these two clusters differentially expressed the Tcm markers CD27 and CCR7 (*Figure 3F*), we then assessed whether Tcm cells were differentially represented among spike-specific CD4+ T cells after each of the vaccination doses. Indeed, Tcm cells were significantly higher after the first dose (*Figure 3G*), consistent with Cluster 8 (enriched after the first dose) expressing high levels of these two receptors. Assessment of other canonical CD4+ T cell subsets – in particular naïve (Tn), stem cell memory (Tscm), effector memory RA (Temra), effector memory (Tem), T transitional memory (Ttm), Tfh, and regulatory T cells (Treg) – revealed Tn cells, like the Tcm subset, to be decreased after the second dose. By contrast, Ttm cells were found to be higher after the second dose, while the remaining subsets were not altered (*Figure 3G-H*). Overall, Tcm and Tfh were the most abundant subsets among the spike-specific CD4+

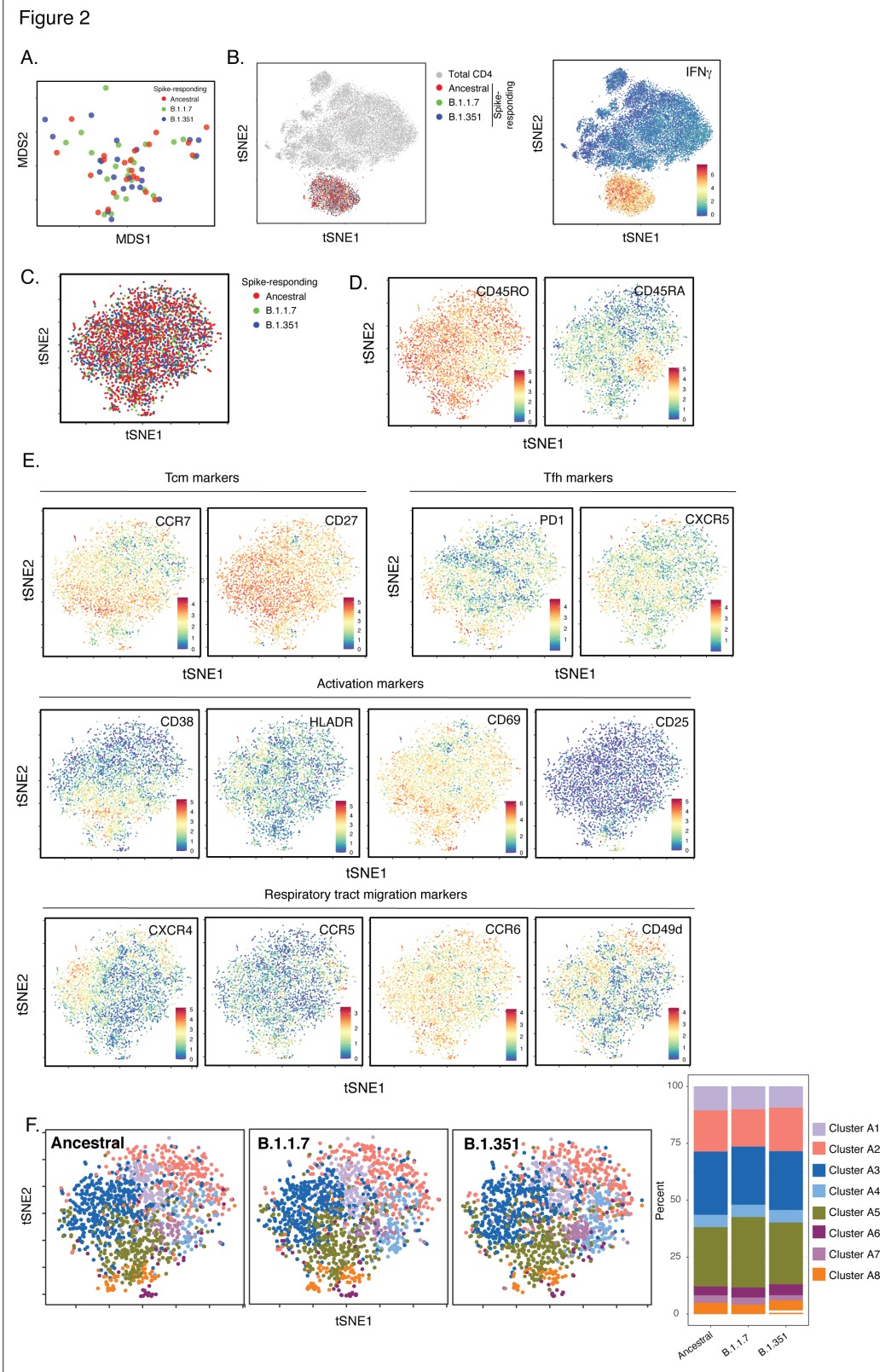

**Figure 2.** SARS-CoV-2-specific CD4+ T cells responding to B.1.1.7 and B.1.351 spike have the same phenotypes as those responding to ancestral spike. (**A**) Datasets corresponding to spike-specific CD4+ T cells after vaccination were visualized as a multidimensional scaling (MDS) plot. Each datapoint reflects the cumulative phenotypes averaged across all the SARS-CoV-2-specific CD4+ T cells from a single stimulated sample. Data for both infection-

*Figure 2 continued on next page*

*Figure 2 continued*

naïve and convalescent individuals, and for both the post-dose one and post-dose two timepoints, are shown. The lack of segregation of the cells responding to the ancestral, B.1.1.7, and B.1.351 spike proteins suggest phenotypic similarities. (**B**) Visualization of the datasets by tSNE dot plots. CD4+ T cells responding to ancestral or variant spike stimulation by producing high amounts of IFNγ (*right*) segregate together and away from the total CD4+ T cell population (*left*). Each dot represents one cell. (**C**) CD4+ T cells responding to ancestral spike and its variants are phenotypically similar, as shown by their complete mingling on a tSNE dot plot. (**D, E**) Spike-responding CD4+ T cells are mostly memory cells, as indicated by high CD45RO and low CD45RA expression levels, and include those expressing high levels of Tcm, Tfh, activation, and respiratory tract migration markers. Shown is the tSNE depicted in *panel C* displaying the relative expression levels of the indicated antigens (Red: high; Blue: low). Heatmaps were scaled from 0 to the maximal signal in each channel. (**F**) CD4+ T cells responding to ancestral spike and its variants distribute in a similar fashion among the eight clusters identified by FlowSOM. Shown on the left is the distribution of T cells responding to ancestral or variant spike peptides on the tSNE depicted in *panel C,* colored according to the FlowSOM clustering. Shown on the right is the quantification of the FlowSOM distribution data. No significant differences were observed between the three groups in the distribution of their cells among the eight clusters, as calculated using a one-way ANOVA and adjusted for multiple testing (n = 8) using Holm-Sidak method (p > 0.05).

The online version of this article includes the following figure supplement(s) for figure 2:

**Figure supplement 1.** Expression levels of all CyTOF phenotyping markers are equivalent between CD4+ T cells responding to stimulation by spike from ancestral, B.1.1.7, and B.1.351 spike.

T cells (*Figure 3G-H*). These data together suggest that after receiving the second dose, infection-naïve individuals' spike-specific CD4+ T cells increase in quantity (*Figure 1B*), and alter their phenotypes as reflected by a decrease Tcm cells and an increase in Ttm cells.

We then conducted a similar analysis in the convalescent individuals. As the pre-vaccination timepoint included spike-specific CD4+ T cells primed by prior SARS-CoV-2 infection, we included all three timepoints in this analysis. When the data were visualized by MDS, it was apparent that most of the pre-vaccination specimens localized away from the post-vaccination specimens, which were interspersed with each other (*Figure 4A*). Similar distinctions between pre-and post-vaccination specimens were visualized at the single-cell level by tSNE, which was particularly apparent when visualized as contour heatmaps (*Figure 4B and C*). Clustering of the cells by FlowSOM revealed that the cluster distribution was markedly skewed among the pre-vaccination cells (*Figure 4D and E*), with one cluster being under-represented (C2) and one over-represented (C5) as compared to both post-vaccination timepoints (*Figure 4F*). Cluster C3 was the only cluster that was significantly different after 1 vs 2 doses (*Figure 4F*) but as this cluster comprised only <5 % of the cells it was not analyzed further. To assess what may drive the differences between the phenotypes of the pre- vs. post-vaccination spike-specific CD4+ T cells, we assessed for markers differentially expressed between clusters C2 and C5. Cluster C2 cells preferentially expressed the Tcm markers CD27 and CCR7, the Tfh markers PD1 and CXCR5, and the co-stimulatory receptors ICOS and Ox40, while among these only CD27 was preferentially expressed in Cluster C5 (*Figure 4—figure supplement 1*). Manual gating confirmed Tcm, Tfh, and ICOS+ Ox40+ cells to be preferentially enriched in the post-vaccination specimens (*Figure 4G-I*). None of the canonical subsets were differentially abundant after the first vs. second vaccination dose. Together, these results suggest that, in contrast to the infection-naïve individuals, convalescents' spike-specific CD4+ T cells were similar after the first vs. second vaccination dose; however, in these individuals vaccination drastically altered the phenotypes of the pre-existing spike-specific CD4+ T cells (presumably elicited from the original infection).

## Vaccination-induced spike-specific CD4+ T cells from convalescent individuals exhibit unique phenotypic features of increased longevity and tissue homing

We next determined whether there were any phenotypic differences between the vaccine-induced spike-specific CD4+ T cells from the infection-naïve vs. convalescent individuals. Removal of convalescent outlier PID4112 revealed the magnitude of the spike-specific CD4+ T cell response to be lower in the convalescents than in infection-naïve participants after full vaccination (*Figure 1D*). But when all donors were included there was no statistically significant difference in response magnitude

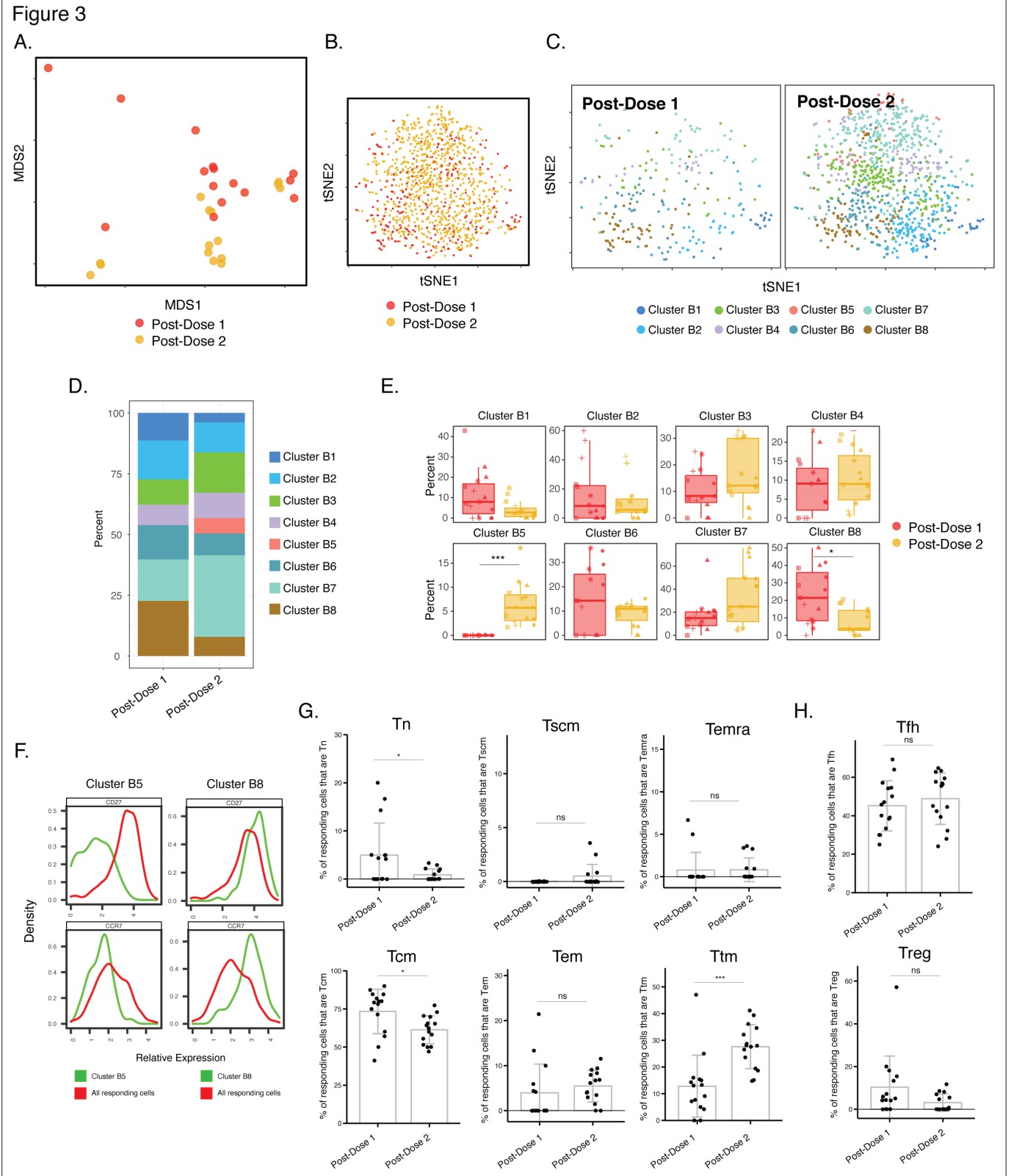

**Figure 3.** Phenotypes of spike-specific CD4+ T cells from infection-naïve individuals following first and second dose of vaccination. (**A**) MDS plot depicting samples of spike-specific CD4+ T cells in vaccinated infection-naïve individuals, showing some interspersion of the cells from the two post-vaccination timepoints. Each dot represents a single specimen. (**B**) tSNE dot plot of spike-specific CD4+ T cells from vaccinated infection-naïve individuals. Each dot represents a single cell. (**C**) tSNE plots depicting cells from the two timepoints, colored according to the cells' cluster classification

*Figure 3 continued on next page*

*Figure 3 continued*

as determined by FlowSOM. (**D**) Distribution among FlowSOM clusters of post-vaccination spike-specific CD4+ T cells from infection-naïve individuals between the two post-vaccination timepoints. (**E**) Two clusters of spike-specific CD4+ T cells (B5 and B8) are differentially abundant after the first vs. second vaccination doses. Data are presented as box plots. *p < 0.05, *** p < 0.001 as determined using student's t-tests adjusted for multiple testing (n = 8) using Holm-Sidak method. (**F**) The Tcm markers CD27 and CCR7 are differentially expressed among Clusters B5 and B8, as depicted by histograms. (**G**) The proportions of Tn (CD45RO-CD45RA + CCR7+ CD95-), Tscm (CD45RO-CD45RA + CCR7+ CD95+), Temra (CD45RO-CD45RA + CCR7-), Tcm (CD45RO + CD45RA-CCR7+ CD27+), Tem (CD45RO + CD45RA-CCR7-CD27-), and Ttm (CD45RO + CD45RA-CCR7-CD27+) among spike-specific CD4+ cells in infection-naïve individuals after the first vs. second vaccination doses. *p < 0.05, ***p < 0.001, ns = non-significant as determined by student's t-test. (**H**) The proportions of Tfh (CD45RO + CD45RA-PD1+ CXCR5+) and Treg (CD45RO + CD45RA-CD25+CD127$^{low}$) among spike-specific CD4+ T cells are similar in infection-naïve individuals after the first vs. second vaccination doses. ns = non-significant as determined by student's t-test. Error bars in panels *G-H* correspond to mean ± SD.

(*Figure 5A*). However, the spike-specific CD4+ T cells from the convalescent and infection-naïve individuals exhibited clear phenotypic differences when assessed by both MDS (*Figure 5B*) and tSNE contours (*Figure 5C*); this was more apparent after the second vaccine dose, but could already be observed after the first. Since the cells after the second dose are more clinically relevant (as they are the ones persisting in vaccinated individuals moving forward), we focused our subsequent analysis on just this timepoint. When visualized as a dot plot, it was apparent that the spike-specific CD4+ T cells from infection-naïve individuals segregated away from those from the convalescents (*Figure 5D*). Clustering of the data also demonstrated differences between the two patient groups (*Figure 5E and F*), which was confirmed by demonstration of a significant difference in Cluster A1 abundance (*Figure 5G*).

To identify these phenotypic differences, we first assessed the relative distributions of the main canonical CD4+ T cell subsets. Interestingly, the vaccinated convalescents harbored significantly more spike-specific Tcm and Tn, and less spike-specific Ttm (*Figure 6A*). By contrast, Tfh and Treg frequencies were not different between infection-naïve and convalescent vaccinees (*Figure 6B*). To broaden our analysis, we assessed for unique features of Cluster A1, which was over-represented in the infection-naïve donors, and Cluster A3, an abundant cluster which was over-represented in the convalescent donors albeit insignificantly (*Figure 5G*). Interestingly, Cluster A1 expressed low levels of CD127, CXCR4, and CCR7 in contrast to Cluster A3 (*Figure 5—figure supplement 1A*). As Cluster A1 is enriched among the infection-naïve individuals, these findings suggest that these three receptors may be expressed at lower levels on the cells from these individuals, relative to those from vaccinated convalescents. This was confirmed by our detection of higher expression of CD127, CXCR4, and CCR7 on spike-specific CD4+ T cells from the convalescents, although for CXCR4 the difference did not reach statistical significance (*Figure 5—figure supplement 1B*).

We then followed up on each of these three differentially expressed markers. CD127, the alpha chain of the IL7 receptor, can drive IL7-mediated homeostatic proliferation of SARS-CoV-2-specific CD4+ T cells *Neidleman et al., 2020b*, and serves as a marker of long-lived precursor memory cells *Kaech et al., 2003*. To assess the potential longevity of the spike-specific CD4+ T cells, we determined the percentage of CD127+ cells expressing low levels of the terminal differentiation marker CD57. After the second dose of vaccination, convalescent individuals harbored more long-lived (CD127+ CD57-) spike-specific CD4+ T cells than infection-naïve individuals (*Figure 6C*). CXCR4, the second preferentially-expressed marker among the convalescents' spike-specific CD4+ T cells, was recently suggested to direct bystander T cells to the lung during COVID-19, and to be co-expressed with the T resident memory / activation marker CD69 *Neidleman et al., 2021*. Interestingly, spike-specific CD4+ T cells from convalescent donors harbored a highly significantly elevated proportion of CXCR4+ CD69+ cells (*Figure 6D*), suggesting a potentially superior ability to migrate into pulmonary tissues. The last differentially expressed antigen, CCR7, is a chemokine receptor that directs immune cells to lymph nodes. As CD62L, a selectin that also mediates lymph node homing, was also on our panel, we assessed whether CCR7+ CD62L+ cells were enriched among the spike-specific CD4+ T cells from the convalescent donors, and found this to be the case (*Figure 6E*).

Our finding that the convalescent donors' spike-specific CD4+ T cells were preferentially CXCR4+ CD69+ and CCR7+ CD62L + suggested that they may preferentially migrate out of the blood into lymphoid tissues. Supporting this possibility was our observation that, after the second vaccine dose, the percentages of CCR7+ CD62L + spike-specific cells increased as the percentages of spike-specific CD4+ T cells decreased (*Figure 6F*). This suggests that the low spike-specific CD4+ T cell response

Figure 4

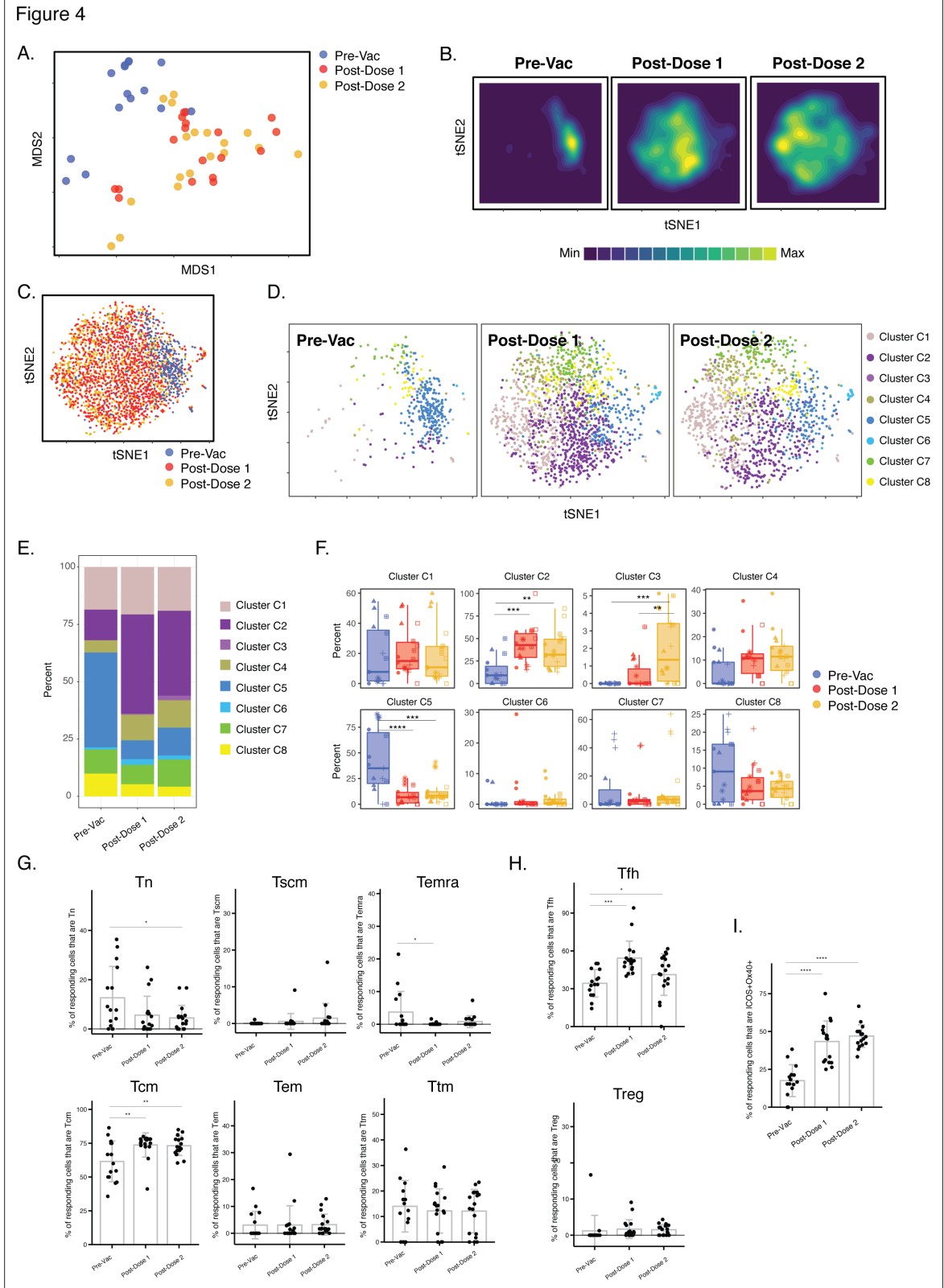

**Figure 4.** Differentiation of spike-specific memory CD4+ T cells after vaccination of convalescent individuals. (**A**) MDS plot depicting datasets corresponding to spike-specific CD4+ T cells in convalescent individuals before and after vaccination. (**B**) tSNE contour heatmaps of spike-specific CD4+ T cells from convalescent individuals emphasizes phenotypic differences between the pre- and post-vaccination cells. Cell densities are represented by color. (**C**) tSNE dot plot of spike-specific CD4+ T cells from convalescent individuals, demonstrating the distinct localization of the pre-vaccination

*Figure 4 continued on next page*

*Figure 4 continued*

cells on the right. (**D**) Spike-specific CD4+ T cells are phenotypically distinct between the pre- and post-vaccination specimens. Shown are tSNE plots depicting cells from the three indicated timepoints, colored according to the cells' cluster classification as determined by FlowSOM. (**E**) The distribution of spike-specific CD4+ T cells classified as FlowSOM clusters differs between the pre- and post-vaccination timepoints. (**F**) Multiple clusters of spike-specific CD4+ T cells are differentially abundant between the pre- and post-vaccination specimens. Data are presented as box plots. **p < 0.01, ***p < 0.001, ****p < 0.0001 as determined by one-way ANOVA and adjusted for multiple testing (n = 8) using the Holm-Sidak method followed by Tukey's honestly significant difference (HSD) post-hoc test. (**G**) Spike-specific CD4+ Tcm increase in convalescent individuals after vaccination. Shown are the proportions of Tn, Tscm, Temra Tcm, Tem, and Ttm among spike-specific CD4+ cells in convalescent individuals before and after vaccination. (**H**) Spike-specific CD4+ Tfh increase in convalescent individuals after vaccination. Shown are the proportions of Tfh and Treg among spike-specific CD4+ T cells in convalescent individuals before and after vaccination. (**I**) Spike-specific CD4+ T cells expressing ICOS and Ox40 increase in convalescent individuals after vaccination. In panels *G-I*, *p < 0.05, **p < 0.01, ***p < 0.001, and ****p < 0.0001 as determined by one-way ANOVA followed by Tukey's HSD post-hoc test. Error bars in panels *G-I* correspond to mean ± SD.

The online version of this article includes the following figure supplement(s) for figure 4:

**Figure supplement 1.** Antigens differentially expressed among Clusters C2 and C5, differentially represented among pre-vs. post-vaccination spike-specific CD4+ T cells from convalescent individuals.

after the second dose of vaccination in some convalescent donors (*Figure 1D*) may have resulted from these cells preferentially leaving the blood compartment. This was further supported by our finding that the expression levels of CCR7 and CD62L on spike-specific CD4+ T cells inversely correlated with the magnitude of the spike-specific CD4+ T cell response (*Figure 6G*). To assess whether the CCR7+ CD62L + and CXCR4+ CD69+ CD4+ T cells have the potential to migrate into the nasopharynx, the most common site of SARS-CoV-2 entry, we obtained paired blood and nasal swabs from one of the participants (PID4101) and phenotyped total CD4+ T cells isolated from these specimens. There was a marked enrichment of both CCR7+ CD62L + and CXCR4+ CD69+ CD4+ T cells in the intranasal specimens (*Figure 6H*), suggesting that CD4+ T cells expressing these markers preferentially exit the blood and enter the nasopharynx. Together, these data suggest that after vaccination, spike-specific CD4+ T cells from convalescent individuals differ from those in infection-naïve individuals in that they appear to be more long-lived, and may more readily migrate out of the blood to mucosal sites, thus explaining their overall lower frequencies measured from the blood.

## Phenotypic features of spike-specific CD8+ T cells from vaccinated, convalescent individuals are unique but differ from their CD4+ T cell counterparts

Finally, we assessed to what extent the main similarities and differences observed with spike-specific CD4+ T cells were also seen for spike-specific CD8+ T cells. Similar to the CD4+ T cells, spike-specific CD8+ T cells stimulated by the three different spike proteins (ancestral, B.1.1.7, B.1.351) did not differ in their phenotypic features (*Figure 6—figure supplement 1A-C*). Also similar to the CD4+ T cells, spike-specific CD8+ T cells elicited by vaccination differed phenotypically in the infection-naïve vs. convalescent individuals (*Figure 6—figure supplement 1D-F*). Unlike the CD4+ T cell data, however, these phenotypic differences could not be accounted for by distribution changes among the main canonical subsets Tn, Tscm, Temra, Tcm, Tem, and Ttm (*Figure 6—figure supplement 1G*). Also unlike the CD4+ T cells, these differences were also not explained by differential abundance of the CD127+ CD57-, CXCR4+ CD69+, or CCR7+ CD62L + subsets (*Figure 6—figure supplement 1H*). Instead, the differences appear to be due to other phenotypic changes, including elevated frequencies of activated cells in the convalescent donors, in particular those co-expressing the Tcm marker CD27 and activation marker CD38, and the checkpoint inhibitor molecule CTLA4 and activation marker 4-1BB (*Figure 6—figure supplement 1I*). These results suggest that vaccine-elicited spike-specific CD8+ T cells, like their CD4+ counterparts, respond equivalently to the B.1.1.7 and B.1.351 variants, and exhibit qualitative differences in convalescent individuals but via different phenotypic alterations than their CD4+ counterparts.

## Discussion

T cells are important orchestrators and effectors during antiviral immunity. They may hold the key to long-term memory due to their ability to persist for decades, yet these cells have been disproportionately

Figure 5

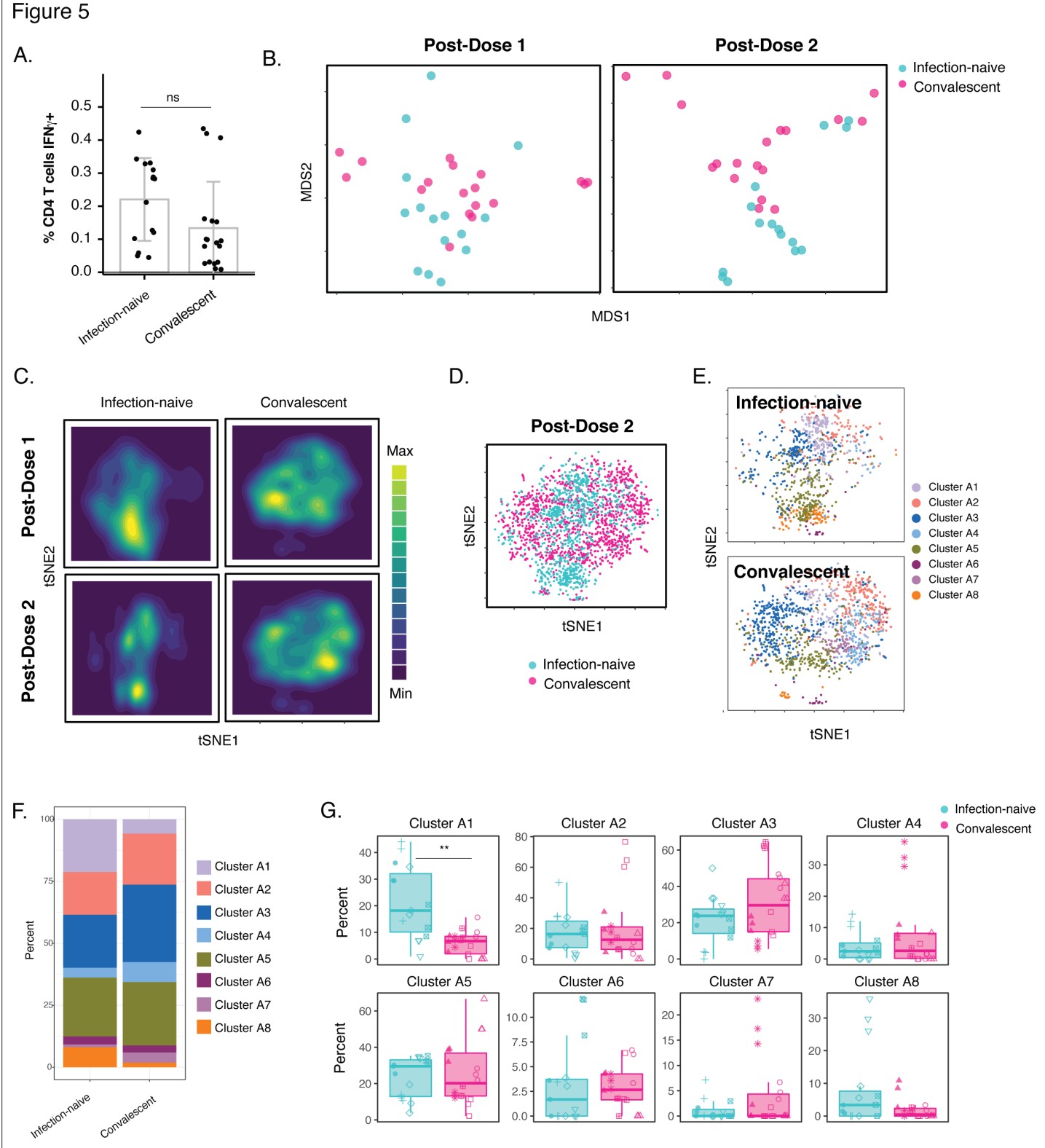

**Figure 5.** Phenotypic features of spike-specific CD4+ T cells differ between infection-naïve and convalescent individuals after vaccination. (**A**) The frequency of spike-specific CD4+ T cells is similar in infection-naïve and convalescent individuals two weeks after the second vaccination dose. Note that when convalescent donor PID4112, who had an unusually high pre-vaccination frequency of spike-specific CD4+ T cells (*Figure 1D*), was excluded, the frequency was significantly lower among the convalescents. Error bars correspond to mean ± SD. (**B**) MDS plots of the phenotypes of spike-specific

*Figure 5 continued on next page*

*Figure 5 continued*

CD4+ T cells in infection-naïve and convalescent individuals after first and second dose vaccinations. (**C**) tSNE contour heatmaps of spike-specific CD4+ T cells from infection-naïve and convalescent individuals, after first and second dose vaccinations, highlighting the phenotypic differences between the two groups of patients. Cell densities are represented by color. (**D**) tSNE dot plot of spike-specific CD4+ T cells from infection-naïve and convalescent individuals after second dose of vaccination, demonstrating the segregation of the cells from the two groups of patients. (**E**) Spike-specific CD4+ T cells are phenotypically distinct between the infection-naïve and convalescent individuals. Shown are tSNE plots depicting cells after the second dose of vaccination, colored according to the cells' cluster classification as determined by FlowSOM. (**F**) The distribution of spike-specific CD4+ T cells into FlowSOM clusters differs between the infection-naïve and convalescent individuals after the second vaccine dose. (**G**) Cluster A1 is over-represented in infection-naïve relative to convalescent individuals after the second dose of vaccination. Data are presented as box plots. **p < 0.01, as determined by student's t-tests adjusted for multiple testing (n = 8) using the Holm-Sidak method.

The online version of this article includes the following figure supplement(s) for figure 5:

**Figure supplement 1.** Cluster A1, enriched among spike-specific CD4+ T cells from infection-naïve relative to convalescent vaccinees, express low levels of markers of homeostatic proliferation and tissue homing.

understudied relative to their humoral immune counterparts in the context of COVID-19. Here, we designed a longitudinal study assessing both the frequency and phenotypic characteristics of SARS-CoV-2-specific T cells in order to address the following questions: (1) Do SARS-CoV-2-specific T cells elicited by vaccination respond similarly to ancestral and variant strains?; (2) To what extent is the second dose needed for boosting T cell responses in infection-naïve and convalescent individuals?; and (3) Do vaccine-elicited memory T cells differ in infection-naïve vs. convalescent individuals?

To answer the first question, we compared post-vaccination SARS-CoV-2 spike-specific T cell responses against ancestral vs. the variant B.1.1.7 and B.1.351 strains. Consistent with recent studies *Skelly et al., 2021*; *Tarke et al., 2021*; *Redd et al., 2021*; *Geers et al., 2021*; *Woldemeskel et al., 2021*; *Stankov et al., 2021*; *Tauzin et al., 2021*, we find that vaccination-elicited T cells specific to the ancestral spike protein also recognize variant spike proteins. We further demonstrate that the phenotypic features of these cells are identical, whether they are stimulated by ancestral or variant spike proteins. This was important to establish because of prior reports that effector T cells can respond differently to APLs by altering their cytokine production or by mounting an immunoregulatory response *Evavold and Allen, 1991*; *Sloan-Lancaster and Allen, 1996*. APLs could theoretically arise when a variant infects an individual that was previously exposed to ancestral spike through vaccination or prior infection. That both the quantity and quality of T cell responses is maintained against the variants may provide an explanation for the real-world efficacy of the vaccines against variants. Although limited data are available, thus far all vaccines deployed in areas where the B.1.1.7 or B.1.351 strains dominate have protected vaccinees from severe and fatal COVID-19 *Gupta, 2021*. Given the potentially important role of SARS-CoV-2-specific T cells in protecting against severe and fatal COVID-19 *Neidleman et al., 2021*; *Dan et al., 2021*, we postulate that this protection may have been in large part mediated by vaccine-elicited T cells. In contrast, efficacy of the vaccines against mild or moderate disease in variant-dominated regions of the world is more variable. For example, in South Africa where B.1.351 is dominant, the AstraZeneca ChAdOx1 vaccine only prevented ~10 % of mild-to-moderate disease cases *Madhi et al., 2021*, while more recent data from Pfizer/BioNTech's vaccine administered in Qatar, where both B.1.1.7 and B.1.351 are dominant, revealed that fully vaccinated individuals were 75 % less likely to develop COVID-19 *Abu-Raddad et al., 2021*. The overall diminished vaccine-mediated protection against milder disease in variant-dominated regions of the world might be explained by the likely important role of antibodies to prevent initial infection by blocking viral entry into host cells (manifesting as protection against asymptomatic and mildly symptomatic infection), and the observation that vaccine-elicited antibodies are generally less effective against the variant than against ancestral spike in lab assays *Wang et al., 2021*; *The CITIID-NIHR BioResource COVID-19 Collaboration et al., 2021*; *Muik et al., 2021*; *Garcia-Beltran et al., 2021*; *Stamatatos et al., 2021*; *Cele et al., 2021*; *Hoffmann et al., 2021*; *Planas et al., 2021*; *Edara et al., 2021*; *Kuzmina et al., 2021*. Reassuringly, there is no evidence that vaccinated individuals mount a weaker immune response to variants than do unvaccinated individuals, which could theoretically result through a phenomenon termed original antigenic sin (where the recall response is inappropriately diverted to the vaccination antigen at the expense of a protective response against the infecting variant strain) *Klenerman and Zinkernagel, 1998*.

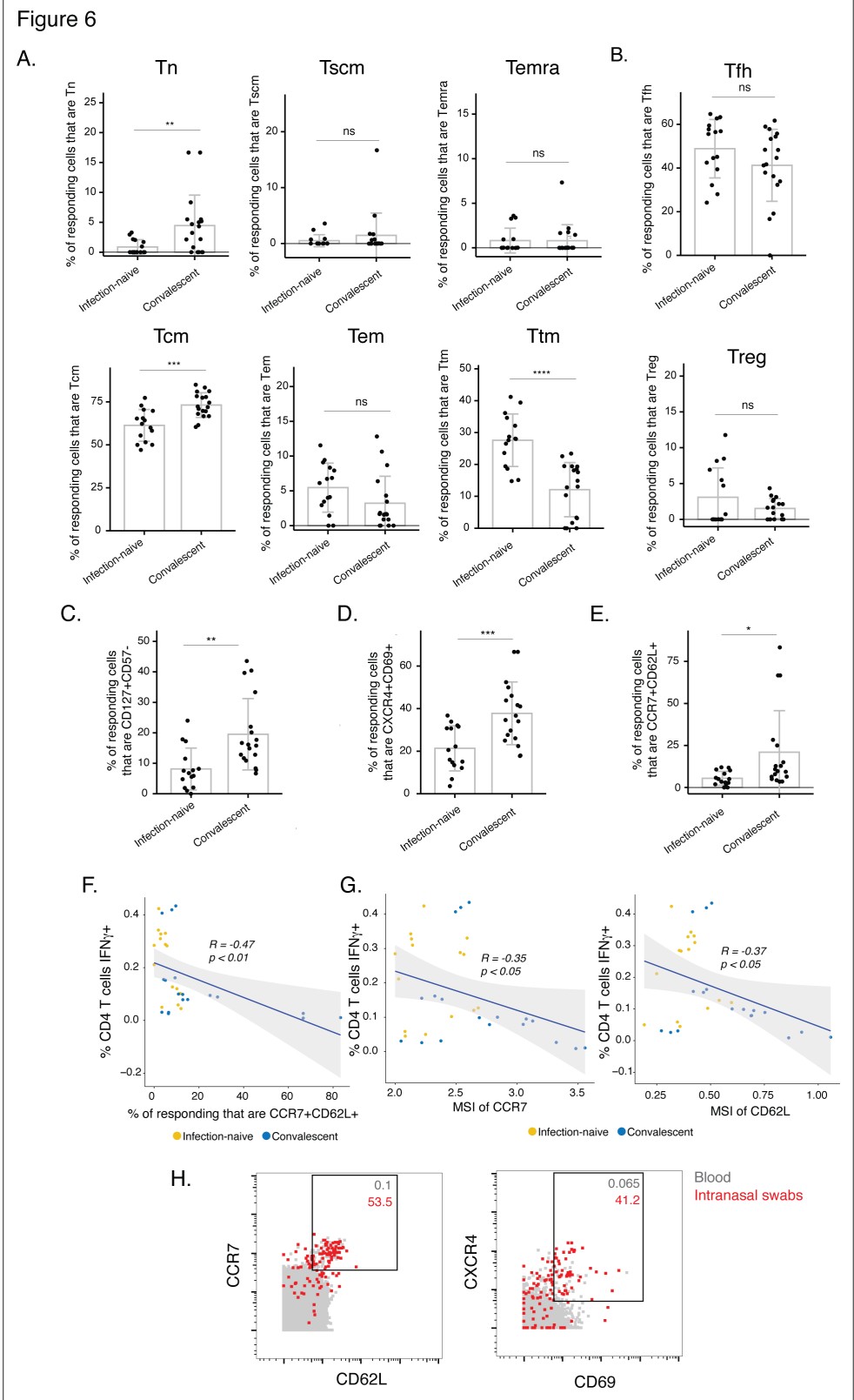

**Figure 6.** The post-vaccination spike-specific CD4+ T cells of convalescents harbor phenotypic features of elevated longevity and tissue homing. (**A**) Spike-specific CD4+ T cells from convalescent vaccinated individuals harbor higher proportions of Tn and Tcm cells and lower proportions of Ttm cells than those from infection-naïve vaccinated individuals. The proportions of Tn, Tscm, Temra, Tcm, Tem, and Ttm cells among spike-specific

*Figure 6 continued on next page*

*Figure 6 continued*

CD4+ T cells were determined by manual gating. **p< 0.01, ***p < 0.001, ****p < 0.0001, ns = non-significant, as determined by student's t-test. (**B**) The proportions of Tfh and Treg among spike-specific CD4+ T cells are similar in infection-naïve vs. convalescent individuals after vaccination. ns = non-significant, as determined by student's t-test. (**C**) Spike-specific CD4+ T cells expressing the homeostatic proliferation marker CD127 and lacking expression of the terminal differentiation marker CD57 are more frequent in vaccinated convalescent than vaccinated infection-naïve individuals. **p < 0.01, as determined by student's t-test. (**D**) Spike-specific CD4+ T cells expressing CXCR4, which directs cells to tissues including the lung, and CD69, a marker of T cell activation and tissue residence, are more frequent in convalescent vaccinated individuals. ***p < 0.001, as determined by student's t-test. (**E**) Spike-specific CD4+ T cells expressing the lymph node homing receptors CCR7 and CD62L are more frequent in vaccinated convalescent individuals. *P < 0.05, as determined by student's t-test. Error bars in panels *A-E* correspond to mean ± SD. (**F**) The proportions of CCR7+ CD62L + cells among spike-specific CD4+ T cells associate negatively with the frequencies of spike-specific CD4+ T cells after the second dose of vaccination (correlation coefficient (**R**) < 0). *P*-values were calculated using t distribution with n-2 degrees of freedom. (**G**) Expression levels (reported as mean signal intensity, or MSI) of CCR7 and CD62L among spike-specific CD4+ T cells associate negatively (*R* < 0) with overall frequencies of spike-specific CD4+ T cells after the second dose of vaccination. p-Values were calculated using t distribution with n-2 degrees of freedom. The 95 % confidence intervals of the regression lines in the scatter plots of *panels F-G* are shaded in grey. (**H**) CCR7+ CD62L + and CXCR4+ CD69+ CD4+ T cells are more frequent in nasopharynx than blood. Unstimulated CD4+ T cells from the blood (*gray*) or from an intranasal swab (*red*) were obtained on the same day from PID4101 and then phenotyped by CyTOF. Numbers indicate the percentages of the corresponding cell population within the gate. Results are gated on live, singlet CD3+ CD4+ CD8- cells.

The online version of this article includes the following figure supplement(s) for figure 6:

**Figure supplement 1.** Phenotypic features of spike-specific CD8+ T cells from vaccinated, convalescent individuals are unique and differ from those of their CD4+ T cell counterparts.

To address the second question of whether a booster dose is needed, we compared the T cells after the first vs. second vaccination doses, among the infection-naïve and convalescent individuals. In infection-naïve individuals, spike-specific responses were observed after the first vaccination dose, and were further boosted after the second. This enhancement of the T cell response after the second dose is similar to the reported increase in anti-spike IgG levels after a second dose in infection-naïve individuals *Goel et al., 2021*; *Ebinger et al., 2021*. Interestingly, phenotypic changes were also observed after the second dose in that the B cells producing the anti-spike antibodies differentiated from IgM-dominant to IgG-dominant producers *Goel et al., 2021*. We also observed some phenotypic changes among spike-specific CD4+ T cells after the second dose, as reflected by an increase in the Ttm response at the expense of the Tcm response. Importantly, however, after either dose, spike-specific CD4+ T cells were still primarily Tcm and Tfh cells, the latter of which are important for providing helper function for B cells. The prominence of SARS-CoV-2-specific Tfh cells after just one dose of vaccination is consistent with prior reports that a single dose of SARS-CoV-2 mRNA in mice is sufficient to elicit potent B and Tfh cell responses in germinal centers *Lederer et al., 2020*. These results suggest that with regard to T cells, the booster dose is necessary for enhancing the magnitude and results in some phenotypic changes although a robust Tfh response is already established the first dose. Overall, our conclusions are in line with those drawn from serological studies *Goel et al., 2021*; *Ebinger et al., 2021*: that it is important to administer the second vaccine dose in infection-naïve individuals to boost spike-specific responses.

A different situation appears to be the case for convalescent individuals. Longitudinal serological studies suggest that the spike-specific antibody response in convalescent individuals after the first mRNA dose is already equivalent to that of infection-naïve individuals after their second mRNA dose *Goel et al., 2021*; *Ebinger et al., 2021*, suggesting that convalescent individuals may only need a single dose of vaccination. We found no evidence of increased numbers of spike-specific CD4+ T cells after the second dose, and minimal phenotypic changes between the cells at the two post-vaccination timepoints. Spike-specific CD4+ T cells from these individuals did however exhibit marked phenotypic changes as they transitioned from the pre- to the post-vaccination timepoints. This was expected since the cells from the pre-vaccination timepoint are resting memory CD4+ T cells that were primed months prior, while the post-vaccination timepoints were more recently-reactivated memory cells. Interestingly, unlike for the infection-naïve individuals where all individuals responded similarly to each

dose of vaccination, the magnitude of the CD4+ T cell response differed markedly between different convalescent individuals. PID4112 had a large pool of spike-specific CD4+ T cells prior to vaccination, and their numbers increased to extremely high levels after the first vaccination dose. Surprisingly, this large peak in the spike-specific response was accompanied by an increase in the nucleocapsid-specific CD4+ T cells, which was unexpected since the vaccine does not contain nucleocapsid. We suspect this high response to nucleocapsid was due to inflammation-mediated bystander activation of T cells in an antigen-independent manner. Consistent with this hypothesis, the participant reported severe side effects (severe headache, chills, myalgia, nausea, and diarrhea) after the first dose. The remaining five convalescent donors, by contrast, never exhibited a robust T cell response, and in fact after full vaccination actually exhibited a highly significantly lower CD4+ T cell response than the infection-naïve vaccinees. We speculate on an explanation further below. Overall, our results suggest that a second SARS-CoV-2 vaccine dose in individuals who have recovered from COVID-19 may provide less benefit than in individuals who have not previously been exposed to SARS-CoV-2; these findings are in line with recommendations from previously published serological studies *Stamatatos et al., 2021*; *Goel et al., 2021*; *Ebinger et al., 2021*.

One of the most striking observations from this study, and the third and final question we set out to answer, was the remarkably distinct phenotypes of spike-specific CD4+ T cells from infection-naïve vs. convalescent individuals who were fully vaccinated. The spike-specific CD4+ T cells from the convalescent individuals harbored features suggesting increased potential for long-term persistence: they were enriched for Tcm cells, which are have longer in vivo half-lives than their Tem and Ttm counterparts *Bacchus-Souffan et al., 2021*, and express elevated levels of CD127, a marker of long-lived memory T cells *Kaech et al., 2003*. Interestingly, CD127 expression on SARS-CoV-2-specific T cells has been implicated in COVID-19 disease amelioration and in these cells' long-term persistence. CD127 expression was more frequent on spike-specific CD4+ T cells from ICU patients who eventually survived severe COVID-19 than in those that did not *Neidleman et al., 2021*. IL7, the ligand for CD127, can drive homeostatic proliferation and expansion of spike-specific CD4+ T cells *Neidleman et al., 2020b*, and CD127 is not only expressed on SARS-CoV-2-specific memory CD4+ and CD8+ T cells, but its levels increase further over the course of convalescence *Neidleman et al., 2020b*; *Ma et al., 2021*. Together, these findings suggest that after vaccination, spike-specific CD4+ T cells in convalescent individuals may persist longer than those from infection-naïve individuals, but additional long-term follow-up studies will be required to directly test whether this indeed is the case.

Another interesting characteristic of post-vaccination spike-specific CD4+ T cells from convalescent individuals relative to infection-naïve individuals was their expression of multiple tissue-homing receptors. In particular, these cells were preferentially CCR7+ CD62L + and CXCR4+ CD69+. CCR7 and CD62L mediate homing to lymph nodes, while CXCR4 is a chemokine receptor important in migration of hematopoietic stem cells to bone marrow, but also able to direct immune cells to the lung during inflammation *Mamazhakypov et al., 2021*. Interestingly, we recently observed co-expression of CXCR4 with CD69 (an activation marker that also identifies T resident memory cells) in pulmonary T cells from COVID-19 patients *Neidleman et al., 2021*. Many of these cells were bystander (non-SARS-CoV-2-specific) CXCR4+ CD69+ T cells whose numbers in blood increased prior to death from COVID-19. We therefore proposed a model whereby recruitment of non-SARS-CoV-2-specific T cells into the lungs of severe patients may exacerbate the cytokine storm and thereby contribute to death *Neidleman et al., 2021*. In the case of the vaccinated convalescent individuals, however, expression of CXCR4 and CD69 on SARS-CoV-2-specific T cells is expected to be beneficial as it would direct the T cells capable of recognizing infected cells into the lung. CCR7 and CD62L co-expression would further enable these cells to enter draining lymph nodes and participate in germinal center reactions. Supporting the hypothesis that the post-vaccination spike-specific CD4+ T cells from convalescent individuals may better home to lymphoid tissues is our observation that frequencies of these cells in blood correlated negatively with the extent to which they co-expressed CCR7 and CD62L. This was further supported by our finding that CD4+ T cells from the nasopharynx of the upper respiratory tract were preferentially CCR7+ CD62L + and CXCR4+ CD69+ relative to their blood counterparts. All together, these results imply that compared to infection-naïve individuals, convalescents' spike-specific CD4+ T cells may be superior in surviving and migrating to the respiratory tract. Directly testing this hypothesis will require obtaining large numbers of respiratory tract cells from vaccinated, infection-naïve vs. convalescent individuals (e.g. via bronchoalveolar lavages or

endotracheal aspirates), or from animal models of SARS-CoV-2 infection, for quantitation and characterization of SARS-CoV-2-specific T cells. Of note, vaccination of infection-naïve individuals might not induce a strong humoral immunity in the respiratory mucosa either, as neutralizing antibodies against SARS-CoV-2 are rarely detected in nasal swabs from vaccinees *Planas et al., 2021*. If it turns out that current vaccination strategies do not ensure robust humoral and cell-mediated immune responses in the respiratory tract, then strategies that better elicit mucosal-homing SARS-CoV-2-specific B and T cells in infection-naïve individuals – for example by implementing an intranasal route of mRNA immunization – may hold a greater chance of achieving sterilizing immunity.

## Limitations

As this study was aimed at using in-depth phenotyping as a discovery tool, it focused on deeply interrogating many different conditions (e.g. spike variants, longitudinal sampling) rather than many donors. Therefore, although a total of 165 CyTOF specimens were run, only 11 donors were analyzed. The main findings reported here should be confirmed in larger cohorts using more cost-effective and high-throughput alternatives to CyTOF such as conventional flow cytometry. Such follow-up studies should also examine the functional outcomes of the discoveries made here (e.g. effect of chemokine receptor expression on homing of infection- and vaccine-elicited SARS-CoV-2-specific T cells), including in animal models of SARS-CoV-2 infection. A second limitation of the study was the need to stimulate the specimens in order to identify and characterize the vaccine-elicited T cells, and our limiting our analyses of SARS-CoV-2-specific T cells to those inducing IFNγ, which may have restricted our ability to characterize subsets such as Tfh cells that are relatively poor producers of this cytokine. We note however that we limited peptide exposure to 6 hours to minimize phenotypic changes caused by the stimulation, similar to our prior studies *Neidleman et al., 2021*; *Neidleman et al., 2020b*. Our analysis focused on CD4+ T cells because the overall numbers of detectable spike-specific CD8+ T cells were low. Nonetheless, the main findings we made with the CD4+ T cells – that they recognize variants equivalently, and that the phenotypes of the responding cells differ by prior SARS-CoV-2 natural infection status – were recapitulated among CD8+ T cells. Future studies should assess the phenotypes of non-stimulated, vaccine-elicited SARS-CoV-2-specific T cells using peptide-MHC tetramers/multimers. Such studies, however, would be limited to analyzing responses against a small number of epitopes, although use of combinatorial tetramers in conjunction with high-parameter phenotyping *Newell et al., 2013* would increase the ability to characterize a larger breadth of the vaccine-elicited T cell response. Such studies however would be limited for the most part to CD8+ T cells as tetramer reagents for CD4+ T cells are less robust. A final limitation is that serological analyses were not performed in this study. As coordination between the humoral and cellular arms of immunity are likely key to effectively controlling viral replication, future studies should assess to what extent the breadth, isotypes, and functional features of spike-specific antibodies elicited by vaccination associate with the herein described phenotypic features of vaccine-elicited SARS-CoV-2-specific T cells.

# Materials and methods

**Key resources table**

| Reagent type (species) or resource | Designation | Source or reference | Identifiers | Additional information |
|---|---|---|---|---|
| Antibody | HLADR (mouse monoclonal) | Thermofisher | Cat#Q22158 | (1 µg/100 µl) |
| Antibody | RORγt (rat monoclonal) | Fisher Scientific | Cat#5013565 | (1 µg/100 µl) |
| Antibody | CD49d (α4) (mouse monoclonal) | Fluidigm | Cat#3141004B | (1 µg/100 µl) |
| Antibody | CTLA4 (mouse monoclonal) | Fisher Scientific | Cat#5012919 | (1 µg/100 µl) |
| Antibody | NFAT (rat monoclonal) | Fluidigm | Cat#3143023 A | (1 µg/100 µl) |

*Continued on next page*

*Continued*

| Antibody | CCR5 (mouse monoclonal) | Fluidigm | Cat#3144007 A | (1 µg/100 µl) |
|---|---|---|---|---|
| Antibody | CD137 (mouse monoclonal) | Fisher Scientific | Cat#BDB555955 | (1 µg/100 µl) |
| Antibody | CD95 (mouse monoclonal) | Fisher Scientific | Cat#MAB326100 | (1 µg/100 µl) |
| Antibody | CD7 (mouse monoclonal) | Fluidigm | Cat#3147006B | (1 µg/100 µl) |
| Antibody | ICOS (hamster monoclonal) | Fluidigm | Cat#3148019B | (1 µg/100 µl) |
| Antibody | Tbet (mouse monoclonal) | Fisher Scientific | Cat#5013190 | (1 µg/100 µl) |
| Antibody | IL4 (rat monoclonal) | Biolegend | Cat#500829 | (1 µg/100 µl) |
| Antibody | CD2 (mouse monoclonal) | Fluidigm | Cat#3151003B | (1 µg/100 µl) |
| Antibody | IL17 (mouse monoclonal) | Biolegend | Cat#512331 | (1 µg/100 µl) |
| Antibody | CD62L (mouse monoclonal) | Fluidigm | Cat#3153004B | (1 µg/100 µl) |
| Antibody | TIGIT (mouse monoclonal) | Fludigm | Cat#3154016B | (1 µg/100 µl) |
| Antibody | CCR6 (mouse monoclonal) | BD Biosciences | Cat#559560 | (1 µg/100 µl) |
| Antibody | IL6 (rat monoclonal) | Biolegend | Cat#501115 | (1 µg/100 µl) |
| Antibody | CD8 (mouse monoclonal) | Biolegend | Cat#301053 | (1 µg/100 µl) |
| Antibody | CD19 (mouse monoclonal) | Biolegend | Cat#302247 | (1 µg/100 µl) |
| Antibody | CD14 (mouse monoclonal) | Biolegend | Cat#301843 | (1 µg/100 µl) |
| Antibody | OX40 (mouse monoclonal) | Fluidigm | Cat#3158012B | (1 µg/100 µl) |
| Antibody | CCR7 (mouse monoclonal) | Fluidigm | Cat#3159003 A | (1 µg/100 µl) |
| Antibody | CD28 (mouse monoclonal) | Fluidigm | Cat#3160003B | (1 µg/100 µl) |
| Antibody | CD45RO (mouse monoclonal) | Biolegend | Cat#304239 | (1 µg/100 µl) |
| Antibody | CD69 (mouse monoclonal) | Fluidigm | Cat#3162001B | (1 µg/100 µl) |
| Antibody | CRTH2 (rat monoclonal) | Fluidigm | Cat#3163003B | (1 µg/100 µl) |
| Antibody | PD-1 (mouse monoclonal) | Biolegend | Cat#329941 | (1 µg/100 µl) |
| Antibody | CD127 (mouse monoclonal) | Fluidigm | Cat#3165008B | (1 µg/100 µl) |
| Antibody | CXCR5 (rat monoclonal) | BD Biosciences | Cat#552032 | (1 µg/100 µl) |

*Continued*

| Antibody | CD27 (mouse monoclonal) | Fluidigm | Cat#3167006B | (1 µg/100 µl) |
|---|---|---|---|---|
| Antibody | IFNγ (mouse monoclonal) | Fluidigm | Cat#3168005B | (1 µg/100 µl) |
| Antibody | CD45RA (mouse monoclonal) | Fluidigm | Cat#3169008B | (1 µg/100 µl) |
| Antibody | CD3 (mouse monoclonal) | Fluidigm | Cat#3170001B | (1 µg/100 µl) |
| Antibody | CD57 (mouse monoclonal) | Biolegend | Cat#359602 | (1 µg/100 µl) |
| Antibody | CD38 (mouse monoclonal) | Fluidigm | Cat#3172007B | (1 µg/100 µl) |
| Antibody | CD4 (mouse monoclonal) | Fluidigm | Cat#3174004B | (1 µg/100 µl) |
| Antibody | CXCR4 (mouse monoclonal) | Fluidigm | Cat#3175001B | (1 µg/100 µl) |
| Antibody | CD25 (mouse monoclonal) | Biolegend | Cat#356102 | (1 µg/100 µl) |

## Human subjects

Eleven participants from the COVID-19 Host Immune Pathogenesis (CHIRP) cohort were recruited for this study. Six were previously infected with SARS-CoV-2 as established by RT-PCR, and had fully recovered from a mild course of disease. Importantly, infections of these six individuals had all occurred in the San Francisco Bay Area between March and July of 2020, when the dominant local strain was the original ancestral (Wuhan) strain. The remaining five participants were not previously infected with the virus. All eleven participants were vaccinated with both doses of either of the Moderna or Pfizer/BioNTech mRNA vaccines (*Table 1*). Blood was drawn from each of the eleven participants prior to vaccination, ~ 2 weeks after the first vaccine dose, and ~2 weeks after the second vaccine dose (33 specimens total). On the day of each blood draw, PBMCs were isolated from blood using Lymphoprep (StemCell Technologies), and then cryopreserved in 90 % fetal bovine serum (FBS) and 10 % DMSO. For participant PID4101, an additional blood-draw and intranasal swab specimens were obtained for immunophenotyping studies. This study was approved by the University of California, San Francisco, and all participants provided informed consent (IRB # 20–30588).

## Preparation of specimens for CyTOF

Cryopreserved PBMCs were revived and cultured overnight to allow for antigen recovery. The cells were then counted, and then 2 million cells per treatment condition were stimulated with the co-stimulatory agents 0.5 µg/ml anti-CD49d clone L25 and 0.5 µg/ml anti-CD28 clone L293 (both from BD Biosciences), in the presence of 0.5 µM of overlapping 15-mer SARS-CoV-2 spike peptides PepMix SARS-CoV-2 peptides from the original SARS-CoV-2 strain, B.1.1.7, or B.1.351, or overlapping 15-mer SARS-CoV-2 nucleocapsid peptides (all from JPT Peptide Technologies). Stimulations were conducted for 6 hours in RP10 media (RPMI 1640 medium (Corning) supplemented with 10 % FBS (VWR), 1 % penicillin (Gibco), and 1 % streptomycin (Gibco)), in the presence of 3 µg/ml Brefeldin A Solution (eBioscience) to enable detection of intracellular cytokines. To establish the phenotypes of total T cells in the absence of stimulation, 2 million cells were cultured in parallel with the stimulated samples, but in the presence of only 3 µg/ml Brefeldin A.

After culture, the cells were treated with cisplatin (Sigma-Aldrich) as a live/dead marker and fixed with paraformaldehyde (PFA) as previously described *Neidleman et al., 2020b*; *Ma et al., 2020*. Cisplatin treatment and fixation was performed as follows: first, cells were resuspended in 2 ml PBS (Rockland) with 2 ml EDTA (Corning), followed by addition of 2 ml PBS/EDTA supplemented with 25 µM cisplatin (Sigma-Aldrich) for 60 s. Cisplatin staining was then quenched with 10 ml of CyFACS (metal contaminant-free PBS (Rockland) supplemented with 0.1 % FBS and 0.1 % sodium azide (Sigma-Aldrich)), centrifuged, and resuspended in 2 % PFA in CyFACS. Fixation was allowed to proceed for

10 minutes at room temperature, after which cells were washed twice with CyFACS, and then resuspended in CyFACS containing 10 % DMSO. Fixed cells were stored at –80 °C until analysis by CyTOF. For paired blood/swab specimens from PID4101, cells were immediately cisplatin-treated and fixed, without prior cryopreservation.

## CyTOF staining and data acquisition

CyTOF staining was conducted in a fashion similar to recently described methods *Neidleman et al., 2021*; *Neidleman et al., 2020b*; *Ma et al., 2020*; *Cavrois et al., 2017*; *Neidleman et al., 2020a*; *Xie et al., 2021*. Cisplatin-treated cells were thawed, counted, and each treatment condition was barcoded using the Cell-ID 20-Plex Pd Barcoding Kit (Fluidigm). After the cells were barcoded and washed, the barcoded samples were combined and diluted to $6 \times 10^6$ cells / 800 µl CyFACS per well in Nunc 96 DeepWell polystyrene plates (Thermo Fisher). Cells were blocked with mouse (Thermo Fisher), rat (Thermo Fisher), and human AB (Sigma-Aldrich) sera for 15 min at 4 °C, and then washed twice in CyFACS. Surface CyTOF antibody staining (*Table 2*) was conducted for 45 min at 4 °C, in a volume of 100 µl / sample. Cells were then washed three times with CyFACS and fixed overnight at 4 °C in 100 µl of 2 % PFA in PBS. The next day, samples were washed twice with Intracellular Fixation & Permeabilization Buffer (eBioscience), and incubated for 45 minutes at 4 °C. After two additional washes with Permeabilization Buffer (eBioscience), samples were blocked for 15 min at 4 °C in 100 µl of Permeabilization Buffer containing mouse and rat sera. After one additional wash with Permeabilization Buffer, samples were stained with the intracellular CyTOF antibodies (*Table 2*) at 4 °C for 45 min in a volume of 100 µl / sample. Cells were then washed once with CyFACS, and stained for 20 min at room temperature with 250 nM of Cell-ID Intercalator-IR (Fluidigm). Immediately prior to sample acquisition, cells were washed twice with CyFACS buffer, once with MaxPar cell staining buffer (Fluidigm), and once with Cell acquisition solution (CAS, Fluidigm). Cells were resuspended in EQ Four Element Calibration Beads (Fluidigm) diluted in CAS immediately prior to acquisition on a Helios-upgraded CyTOF2 instrument (Fluidigm) at the UCSF Parnassus flow core facility.

## CyTOF data analysis

CyTOF datasets, exported as flow cytometry standard (FCS) files, were de-barcoded and normalized according to manufacturer's instructions (Fluidigm). FlowJo software (BD Biosciences) was used to identify CD4+ T cells (live, singlet CD3+ CD19 CD4+ CD8-) and CD8+ T cells (live, singlet CD3+ CD19 CD4-CD8+) among all analyzed samples. IFNγ+ in the stimulated samples were considered to be the SARS-CoV-2-responsive cells. For high-dimensional analyses of SARS-CoV-2-specific T cells among the stimulated samples, we excluded samples with an insufficient number of events ( ≤ 3) to limit skewing of the data. Manual gating analysis was initially performed using FlowJo, and then select populations were exported as FCS files and then imported into R software as GatingSet objects. Using the *CytoExploreR* package, 2D-gates were manually drawn on the imported samples. The 2D dot plots and statistical results were exported for data visualization, bar-graph generation, and statistical comparisons as previously described (https://github.com/DillonHammill/CytoExploreR; *Hammill, 2021*). High-dimensional analyses (MDS, tSNE, and FlowSOM) were performed using R software by implementing a CyTOF workflow recently described *Nowicka et al., 2017*.

For MDS plot generation, we used the plotMDS function from the *limma* package with default settings. Euclidean distances between all samples were calculated using the arcsinh-transformed median expression levels with cofactor 5, of the lineage and functional markers listed below.

| CD8 | Lineage (Only for CD8 subset) |
|---|---|
| CD4 | Lineage (Only for CD4 subset) |
| CD161 | Lineage |
| HLADR | Lineage |
| CD45RO | Lineage |

*Continued on next page*

*Continued*

| CD8 | Lineage (Only for CD8 subset) |
|---|---|
| CD69 | Lineage |
| CRTH2 | Lineage |
| PD1 | Lineage |
| CXCR5 | Lineage |
| CD27 | Lineage |
| CD3 | Lineage |
| CD2 | Lineage |
| CD62L | Lineage |
| CCR6 | Lineage |
| OX40 | Lineage |
| CD28 | Lineage |
| CD127 | Lineage |
| RORγt | Lineage |
| CXCR4 | Lineage |
| CTLA4 | Lineage |
| NFAT | Lineage |
| CCR5 | Lineage |
| CD137 | Lineage |
| CD95 | Lineage |
| ICOS | Lineage |
| CD49d | Lineage |
| CD7 | Lineage |
| Tbet | Lineage |
| TIGIT | Lineage |
| CCR7 | Lineage |
| CD45RA | Lineage |
| CD57 | Lineage |
| CD38 | Lineage |
| α4β7 | Lineage |
| CD25 | Lineage |
| IFNγ | Function |
| IL6 | Function |
| IL4 | Function |
| IL17 | Function |

The first (MDS1) and second (MDS2) MDS dimensions were plotted to show the dissimilarities between samples from the indicated conditions as described *Ritchie et al., 2015*.

tSNE was performed using the Trsne function from the *Rtsne* package using arcsinh-transformed expression of lineage markers (no PCA step, iterations = 1000,, perplexity = 30, theta = 0.5). Events

corresponding to unstimulated T cells were down-sampled to 1000 cells per sample, and SARS-CoV-2-specific cells (cell numbers ranging from 4 to 229 per sample) were all included in the tSNE analyses without down-sampling. Each cell was displayed in a tSNE plot for dimension reduction visualization and colored with arcsinh-transformed cell marker expression as heatmaps, or pseudo-colored by the appropriate group.

Unsupervised cell subset clustering was performed using FlowSOM *Van Gassen et al., 2015* and *ConsensusClusterPlus* packages using arcsinh-transformed expression levels of the lineage markers indicated above *Wilkerson and Hayes, 2010*. For clustering of SARS-CoV-2-specific T cells, we set the meta-cluster number to eight and cluster number to 40. The frequency of each cluster within each sample was calculated using the following equation:

(Frequency of cluster in specified sample) = (Cell count of cluster / Total cell count of specified sample).

This was then converted to a percentage by multiplying by 100. The percentages of each cluster from the selected samples were plotted as box plots with jittered points, followed by statistical analysis between the groups. To compare the abundance distribution of clusters between groups, frequencies of clusters in samples from each group were normalized using the equation below:

(Normalized frequency of cluster in specified sample) = (Frequency of cluster in specified sample/ Total number of samples in each group).

This was then converted to a percentage by multiplying by 100, and plotted as stacked bar charts.

## Statistical analysis

The statistical tests used in comparison of groups are indicated within the figure legends. For 2-group comparisons, student's t-tests were performed and p-values were adjusted for multiple testing using the Holm-Sidak method where applicable. For comparisons of three or more groups, significance between groups was first evaluated by one-way ANOVA, and then the p-values were adjusted for multiple testing using the Holm-Sidak method where applicable. For datasets with significant ANOVA-adjusted p-values ( $\leq 0.05$ ), we performed Tukey's honestly significant difference (HSD) post-hoc test to determine the p-values between individual groups.

## Raw data availability

For this study, a total of 120 specimens were analyzed by CyTOF. Each specimen included both CD4+ and CD8+ T cells. For each specimen, we gated separately on events corresponding to CD4+ T cells (live, singlet CD3+ CD19- CD4+ CD8-) and CD8+ T cells (live, singlet CD3+ CD19- CD4 CD8+), and exported the files as 240 individual FCS files. These 240 raw CyTOF datasets are available for download through the public repository Dryad via the following link: https://doi.org/10.7272/Q60R9MMK.

## Acknowledgements

This work was supported by the Van Auken Private Foundation, David Henke, and Pamela and Edward Taft (N.R.R.); philanthropic funds donated to Gladstone Institutes by The Roddenberry Foundation and individual donors devoted to COVID-19 research (N.R.R.); the Program for Breakthrough Biomedical Research (N.R.R., S.A.L.), which is partly funded by the Sandler Foundation; and Awards #2164 (N.R.R.), #2208 (N.R.R.), and #2160 (to S.A.L.) from Fast Grants, a part of Emergent Ventures at the Mercatus Center, George Mason University. We acknowledge the NIH DRC Center Grant P30 DK063720 and the S10 1S10OD018040-01 for use of the CyTOF instrument. We thank Stanley Tamaki and Claudia Bispo for CyTOF assistance at the Parnassus Flow Core, Heather Hartig for help with recruitment, Françoise Chanut for editorial assistance, and Robin Givens for administrative assistance.

## Additional information

### Funding

| Funder | Grant reference number | Author |
| --- | --- | --- |
| Sandler Foundation | Program for Breakthrough Biomedical Research | Nadia R Roan Sulggi A Lee |
| Fast Grants | 2164 | Nadia R Roan |
| Fast Grants | 2208 | Nadia R Roan |
| Fast Grants | 2160 | Sulggi A Lee |

The funders had no role in study design, data collection and interpretation, or the decision to submit the work for publication.

### Author contributions

Jason Neidleman, Data curation, Formal analysis, Investigation, Methodology, Writing - review and editing; Xiaoyu Luo, Data curation, Formal analysis, Methodology, Visualization, Writing - review and editing; Matthew McGregor, Guorui Xie, Investigation; Victoria Murray, Resources; Warner C Greene, Funding acquisition, Supervision, Writing - review and editing; Sulggi A Lee, Conceptualization, Funding acquisition, Project administration, Resources, Supervision, Writing - review and editing; Nadia R Roan, Conceptualization, Data curation, Formal analysis, Funding acquisition, Methodology, Project administration, Resources, Supervision, Writing - original draft

### Author ORCIDs

Warner C Greene http://orcid.org/0000-0001-9896-8615
Sulggi A Lee http://orcid.org/0000-0003-1560-2250
Nadia R Roan http://orcid.org/0000-0002-5464-1976

### Ethics

Human subjects: This study was approved by the University of California, San Francisco (IRB # 20-30588). All participants provided informed consent, and consent to publish, before participation.

### Decision letter and Author response

Decision letter https://doi.org/10.7554/eLife.72619.sa1
Author response https://doi.org/10.7554/eLife.72619.sa2

## Additional files

### Supplementary files

• Transparent reporting form

### Data availability

The original datasets are available through Dryad: https://doi.org/10.7272/Q60R9MMK.

The following dataset was generated:

| Author(s) | Year | Dataset title | Dataset URL | Database and Identifier |
| --- | --- | --- | --- | --- |
| Roan NR | 2021 | mRNA vaccine-induced T cells respond identically to SARS-CoV-2 variants of concern but differ in longevity and homing properties depending on prior infection status | https://doi.org/10.7272/Q60R9MMK | Dryad Digital Repository, 10.7272/Q60R9MMK |

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
