## [Decision Letter]

**Acceptance summary:**

This work will be of broad interest to those studying adaptive immunity to SARS-CoV-2, particularly with a focus on T cell immunology. The study confirms that mRNA vaccine-elicited T cell responses maintain recognition of peptides derived from VOC, and identifies several phenotypic differences in the T cell response elicited by infection, vaccination, or vaccination following infection. The insights generated by this work will inform further study of SARS-CoV-2-specific T cell responses and their role in supporting protective immunity.

**Decision letter after peer review:**

Thank you for submitting your article "mRNA vaccine-induced T cells respond identically to SARS-CoV-2 variants of concern but differ in longevity and homing properties depending on prior infection status" for consideration by *eLife*. Your article has been reviewed by 2 peer reviewers, and the evaluation has been overseen by a Reviewing Editor and Miles Davenport as the Senior Editor. The following individual involved in review of your submission has agreed to reveal their identity: Andrew J McMichael (Reviewer #2).

Essential revisions:

1) Please respond to questions 1 and 2 from reviewer #1, in particular modifying the discussion to address question 1.

2) If there is an opportunity to include any serological data on the response to vaccination among the cohorts included in the study, this would strengthen the paper, as mentioned by both reviewers.

3) Reviewer #2 has highlighted that, as an observational study, the discussion could be strengthened by acknowledgement of how future studies could identify the protective roles of the T cell phenotypes identified in this work.

*Reviewer #1 (Recommendations for the authors):*

In general, the manuscript is clearly written, the experiments are performed well, and the conclusions are supported by the data. My comments are largely restricted to modifications of the text that can improve the interpretation of the results and discussion of some of the limitations of the study.

1) The authors have chosen to base their identification of spike-specific T cells on IFNγ secretion, which is understandable. However, given that cTfh cells, in particular, are known to be poor producers of IFNγ, and also the fact that CD4 T cells may express CD40L or other cytokines in the absence of IFNγ, it would be useful to briefly discuss the impact this may have on the results

2) Figure 2E appears to show high CCR6 expression on nearly all of the spike-specific T cells – given that these cells were identified by IFNγ (typically associated with Th1/CXCR3 expression), is it surprising that these cells are enriched for CCR6 expression?

3) Do the authors have serological data from the participants in this study? It would be helpful to confirm whether, similar to other cohorts, the vaccinated convalescent individuals exhibited high titres of neutralising antibodies, and whether there was any association between serological outcomes and the CD4 T cell responses described.

*Reviewer #2 (Recommendations for the authors):*

The study raises the question of how these differences affect susceptibility to infection and disease severity. How age impacts on these results is also a further question. However answers to these questions cannot be obtained in small groups such as these and within a study taking months rather than years. Therefore I think it is not reasonable to ask for these additions but instead to encourage the authors too become involved in such studies in the future.

What they have presented is detailed and thorough and they have demonstrated some clear differences between their vaccinated donor groups. However the groups are small 5 vs 6, and one of the 6 was an outlier. Expanding the group sizes would increase confidence.

Overall the study is very descriptive which is not necessarily a negative thing but it does raise a lot of questions as to what it all means and which of these T cell subsets studied are protective and how.

---

## [Author Response]

Essential revisions:1) Please respond to questions 1 and 2 from reviewer #1, in particular modifying the discussion to address question 1.

We have responded to these questions further below, and have modified the discussion to address question 1 from Reviewer #1.

2) If there is an opportunity to include any serological data on the response to vaccination among the cohorts included in the study, this would strengthen the paper, as mentioned by both reviewers.

Serological assays were not performed in this study; however we fully agree with the importance of associating the in-depth phenotypes of vaccine-elicited SARS-CoV-2-specific T cells with the antibody response. In fact, just as we went very “deep” into the phenotypes of SARS-CoV-2-specific T cells in this study, we are at the moment optimizing techniques to, in an analogous fashion, deeply characterize the serological response to vaccination. This entails optimizing a flow cytometry-based approach we recently introduced and implemented on a small number of specimens (Ma et al., J Immunol 207(5):1344, PMID 34389625), to be able to simultaneously assess the levels of IgA1, IgA2, IgE, IgG1, IgG2, IgG3, IgG4, and IgM against the S1, S2, and RBD domains of the SARS-CoV-2 spike protein in a large number of patient specimens. Once we’ve optimized the assay and applied it on the vaccine specimens, we plan to associate the resulting 24-parameter serological datasets (8 isotypes of antibodies each against 3 antigens = 24 parameters total) with the high-dimensional SARS-CoV-2-specific T cell datasets from this study, but that will be its own separate (and large) study and beyond the scope of this current one. As generating such serological data will take at least 3-6 months to complete, and the focus of this study is on SARS-CoV-2-specific T cells (and all conclusions we drew were based only on the T cell data), we think it appropriate that we limit this study to deep-phenotyping of the T cells. We have now brought up in the last part of our “Limitations” section the lack of serological analysis in this current study as a limitation, and how follow-up studies should associate serological responses with the T cell responses characterized here.

Lines 506-511: “A final limitation is that serological analyses were not performed in this study. As coordination between the humoral and cellular arms of immunity are likely key to effectively controlling viral replication, future studies should assess to what extent the breadth, isotypes, and functional features of spike-specific antibodies elicited by vaccination associate with the herein described phenotypic features of vaccine-elicited SARS-CoV-2-specific T cells.”

3) Reviewer #2 has highlighted that, as an observational study, the discussion could be strengthened by acknowledgement of how future studies could identify the protective roles of the T cell phenotypes identified in this work.

We have addressed this point by adding to the discussion how future studies should define the protective roles of the T cells characterized in this study, and that such studies will require animal models (Lines 488-491: “Such follow-up studies should also examine the functional outcomes of the discoveries made here (e.g., effect of chemokine receptor expression on homing of infection- and vaccine-elicited SARS-CoV-2-specific T cells), including in animal models of SARS-CoV-2 infection.”).

Reviewer #1 (Recommendations for the authors):In general, the manuscript is clearly written, the experiments are performed well, and the conclusions are supported by the data.

We thank the reviewer for the positive comments.

My comments are largely restricted to modifications of the text that can improve the interpretation of the results and discussion of some of the limitations of the study.1) The authors have chosen to base their identification of spike-specific T cells on IFNγ secretion, which is understandable. However, given that cTfh cells, in particular, are known to be poor producers of IFNγ, and also the fact that CD4 T cells may express CD40L or other cytokines in the absence of IFNγ, it would be useful to briefly discuss the impact this may have on the results.

In fact our 39-parameter CyTOF panel theoretically allowed us to characterize not only SARS-CoV-2-specific CD4^+^ T cells producing IFNg, but also those producing other canonical effector cytokines of CD4^+^ T cells (IL4 for Th2 cells, IL17 for Th17 cells); however, we did not detect any SARS-CoV-2-specific IL4+ or IL17+ cells, consistent with our prior studies from COVID-19 patients (Neidleman et al., Cell Reports Medicine 2020 1(6):100081 PMID: 32839763; Neidleman et al., Cell Reports 2021 36(3):109414 PMID: 34260965). We have now added to Figure S1 the data showing our inability to find any vaccine-elicited SARS-CoV-2-specific Th2 and Th17 cells.

We also note that although CD40L was not on our panel, other activation markers typically used in AIM assays (Ox40, 4-1BB, and CD69) were, and as shown in Figure S1 of our original manuscript, could not be used to specifically identify SARS-CoV-2-specific T cells as the background (AIM+ cells detected when no peptide was added) was too high. Although stimulating with peptide for 24 instead of 6 hours would be expected to increase the ability to specifically detect SARS-CoV-2-specific T cells, such a long period of stimulation would also be expected to more drastically alter the phenotypes of the cells from their original baseline state, which we did not want for our phenotyping studies.

To acknowledge the limitation of our restricting our analyses to IFNg+ responding cells, we have now added to our “Limitations” section our having limited “our analyses of SARS-CoV-2-specific T cells to those inducing IFNg, which may have restricted our ability to characterize subsets such as Tfh cells that are relatively poor producers of this cytokine” (Lines 491-494). In addition, we have commented that future studies should take advantage of tetramer technology (which would enable analysis beyond just the IFNg-responding ones), whilst acknowledging that such studies would for the most part be limited to CD8^+^ T cell responses as tetramer reagents for CD4^+^ T cells are less robust (Lines 500-506).

2) Figure 2E appears to show high CCR6 expression on nearly all of the spike-specific T cells – given that these cells were identified by IFNγ (typically associated with Th1/CXCR3 expression), is it surprising that these cells are enriched for CCR6 expression?

We apologize for the confusion, in fact the heatmaps in Figure 2E were scaled from 0 to the maximal signal within each channel. The overall yellow coloring of the CCR6 heatmap did not mean that all the cells expressed high levels of CCR6, but rather that the dynamic range of CCR6 expression was low relative to other antigens, and most cells shown in the Figure 2E heatmap in fact expressed intermediate levels of CCR6.

To illustrate this point, we have shown within Author response image 1 tSNE heatmaps corresponding to total CD4^+^ T cells from the specimens. This tSNE space was depicted in Figure 2B of the original manuscript. As shown in panel A, the “island” at the bottom corresponds to the IFNg+ (SARS-CoV-2-specific) T cells in the stimulated sample; that these cells are mostly Th1 cells is confirmed by their preferentially expressing the Th1 transcription factor Tbet (right side of Author response image 1). Panel B shows CCR6 expression levels in the same tSNE space, where it’s apparent that CCR6 expression is not especially high among the SARS-CoV-2-specific T cells, but rather somewhat uniform throughout the entire CD4^+^ T cell population. By contrast, as shown in panel C, two other chemokine receptors (CXCR4 and CCR5) show a much larger dynamic range, where there are were clearer distinctions between the low (blue) and high (red) expressors of these antigens.

**Author response image 1. sa2fig1:** 

Consistent with CCR6 being relatively “uniform” among all the CD4^+^ T cells was our visualization of CCR6 expression levels as 2D dot plots, where it’s apparent that the highest expressers of CCR6 were in fact not IFNg+ but were IFNg^low^ and infrequent (region highlighted by red arrow within Author response image 2).

In conclusion, it was not the case that the IFNg+ SARS-CoV-2-specific T cells were enriched for CCR6 expression, but rather that they expressed moderate levels of CCR6 comparable to that expressed by other CD4^+^ T cells. We have now clarified within the legend of Figure 2 (Lines 708-709) that each heatmap was scaled from 0 to the maximal signal within each channel.

3) Do the authors have serological data from the participants in this study? It would be helpful to confirm whether, similar to other cohorts, the vaccinated convalescent individuals exhibited high titres of neutralising antibodies, and whether there was any association between serological outcomes and the CD4 T cell responses described.

Serological assays were not performed in this study; however we fully agree with the importance of associating the in-depth phenotypes of vaccine-elicited SARS-CoV-2-specific T cells with the antibody response. In fact, just as we went very “deep” into the phenotypes of SARS-CoV-2-specific T cells in this study, we are at the moment optimizing techniques to, in an analogous fashion, deeply characterize the serological response to vaccination. This entails optimizing a flow cytometry-based approach we recently introduced and implemented on a small number of specimens (Ma et al., J Immunol 207(5):1344, PMID 34389625), to be able to simultaneously assess the levels of IgA1, IgA2, IgE, IgG1, IgG2, IgG3, IgG4, and IgM against the S1, S2, and RBD domains of the SARS-CoV-2 spike protein in a large number of patient specimens. Once we’ve optimized the assay and applied it on the vaccine specimens, we plan to associate the resulting 24-parameter serological datasets (8 isotypes of antibodies each against 3 antigens = 24 parameters total) with the high-dimensional SARS-CoV-2-specific T cell datasets from this study, but that will be its own separate (and large) study and beyond the scope of this current one. As generating such serological data will take at least 3-6 months to complete, and the focus of this study is on SARS-CoV-2-specific T cells (and all conclusions we drew were based only on the T cell data), we think it appropriate that we limit this study to deep-phenotyping of the T cells. We have now brought up in the last part of our “Limitations” section the lack of serological analysis in this current study as a limitation, and how follow-up studies should associate serological responses with the T cell responses characterized here. (Lines 506-511: “A final limitation is that serological analyses were not performed in this study. As coordination between the humoral and cellular arms of immunity are likely key to effectively controlling viral replication, future studies should assess to what extent the breadth, isotypes, and functional features of spike-specific antibodies elicited by vaccination associate with the herein described phenotypic features of vaccine-elicited SARS-CoV-2-specific T cells.”)

Reviewer #2 (Recommendations for the authors):The study raises the question of how these differences affect susceptibility to infection and disease severity. How age impacts on these results is also a further question. However answers to these questions cannot be obtained in small groups such as these and within a study taking months rather than years. Therefore I think it is not reasonable to ask for these additions but instead to encourage the authors too become involved in such studies in the future.What they have presented is detailed and thorough and they have demonstrated some clear differences between their vaccinated donor groups. However the groups are small 5 vs 6, and one of the 6 was an outlier. Expanding the group sizes would increase confidence.Overall the study is very descriptive which is not necessarily a negative thing but it does raise a lot of questions as to what it all means and which of these T cell subsets studied are protective and how.

We thank the Reviewer for bringing up these important points, and for recognizing that we had implemented CyTOF as a discovery tool, and that the discoveries made here will need to be validated, in ways that account for the impact of age and how the SARS-CoV-2-specific T cell features defined herein affect susceptibility to infection and disease severity. Such studies will most certainly necessitate not only larger cohorts, but also studies in animal models. We in fact are planning pushing forward on all these fronts. We are collecting patient specimens from additional vaccinated patients (including those with breakthrough infections) to increase our “n”; we have initiated studies in a mouse model of pathogenic SARS-CoV-2 infection; and we have upcoming plans (after having received a pilot grant) to characterize vaccine-elicited tissue SARS-CoV-2-specific T cell responses in non-human primates. To highlight the importance of these future studies, we have now in the revised version of the manuscript highlighted the need to perform validation and functional studies in additional donors and in animal models (Lines 486-491: “The main findings reported here should be confirmed in larger cohorts using more cost-effective and high-throughput alternatives to CyTOF such as conventional flow cytometry. Such follow-up studies should also examine the functional outcomes of the discoveries made here (e.g., effect of chemokine receptor expression on homing of infection- and vaccine-elicited SARS-CoV-2-specific T cells), including in animal models of SARS-CoV-2 infection”).